# Jena Soil Model (JSM v1.0; revision 1934): A microbial soil organic carbon model integrated with nitrogen and phosphorus processes

Lin Yu[1], Bernhard Ahrens[1], Thomas Wutzler[1], Marion Schrumpf[1,2], and Sönke Zaehle[1,2]

[1]Max Planck Institute for Biogeochemistry, Hans-Knöll Str. 10, 07745 Jena, Germany
[2]International Max Planck Research School (IMPRS) for Global Biogeochemical Cycles, Jena, Germany

**Correspondence:** Lin Yu <lyu@bgc-jena.mpg.de>

**Abstract.** Plant–soil interactions, such as the coupling of plants' below-ground biomass allocation with soil organic matter (SOM) decomposition, nutrient release and plant uptake, are essential to understand the response of carbon (C) cycling to global changes. However, these processes are poorly represented in the current terrestrial biosphere models owing to the simple first-order approach of SOM cycling and the ignorance of variations within a soil profile. While the emerging microbially explicit soil organic C models can better describe C formation and turnover, at present, they lack a full coupling to the nitrogen (N) and phosphorus (P) cycles with the soil profile. Here we present a new SOM model—the Jena Soil Model (JSM)—which is microbially explicit, vertically resolved and integrated with the N and P cycles. To account for the effects of nutrient availability and litter quality on decomposition, JSM includes the representation of enzyme allocation to different depolymerisation sources based on the microbial adaptation approach as well as of nutrient acquisition competition based on the equilibrium chemistry approximation approach. Herein, we present the model structure and basic features of model performance in a beech forest in Germany. The model reproduced the main SOM stocks and microbial biomass as well as their vertical patterns in the soil profile. We further tested the sensitivity of the model to parameterisation and showed that JSM is generally sensitive to changes in microbial stoichiometry and processes.

## 1 Introduction

There is ample evidence from both ecosystem monitoring data (Bond-Lamberty et al., 2018; Hou et al., 2018; Jonard et al., 2015) and ecosystem manipulation experiments (Ellsworth et al., 2017; Iversen et al., 2012; McCarthy et al., 2010; Warren et al., 2011) that the effects of environmental changes, such as atmospheric $CO_2$ concentrations, global warming and continued air pollution, on terrestrial ecosystems are driven by the constraints imposed by macronutrients such as nitrogen (N) and phosphorus (P). It is, therefore, of great relevance to identify and understand these constraints on global carbon (C) cycling and storage for predicting potential future carbon climate feedback (Ciais et al., 2013). There have been continuous efforts to integrate the N (Thornton et al., 2007; Zaehle and Friend, 2010; Smith et al., 2014) and P cycles (Wang et al., 2010; Yang et al., 2014b; Goll et al., 2017; Thum et al., 2019; Zhu et al., 2019) in terrestrial biosphere models (TBMs) for improving the representation of C-nutrient interactions. However, despite major advances in simulating terrestrial biogeochemistry, these nutrient-enabled TBMs largely fail to reproduce the responses of ecosystems to elevated atmospheric $CO_2$ concentration, as

observed in the free air $CO_2$ enrichment experiments (Zaehle et al., 2014; Medlyn et al., 2015, 2016; Fleischer et al., 2019). An important shortcoming of the current generation of models is their poor representation of plant–soil interactions, in particular the responses of soil organic matter (SOM) decomposition and nutrient release to altered plant inputs and ultimately plant uptake of mineral nutrients (Hinsinger et al., 2011; Drake et al., 2011; Zaehle et al., 2014).

Current TBMs largely adopt the CENTURY model (Parton et al., 1988) or comparable model approaches, in which SOM is divided into two or three pools with different first-order decomposition rates. In these models, nutrient mineralisation and immobilisation fluxes depend on the C transfer efficiency between SOM pools and their prescribed C:N:P stoichiometry. Recent insights in soil science have questioned the adequacy of the CENTURY approach to SOM cycling for simulating the effects of global changes, particularly in response to altered plant inputs. Researchers underscored the need and offered a direction for a
more mechanistic representation of soil processes in models, such as the substrate limitation of soil microbial growth as well as the nutrient immobilisation and physical-chemical stabilisation of organic matter through organo-mineral association (Schmidt et al., 2011; Lehmann and Kleber, 2015). Another limitation of many current SOM models in TBMs is that they represent soil as a 'bucket', thus ignoring the strong variance of SOM cycling within a soil profile (Koven et al., 2013; Arora et al., 2013; McGuire et al., 2018). Such a highly empirical representation of SOM cycling, in which important processes such as microbial
immobilisation or rhizosphere deposition are not well represented, brings large uncertainties in future projections of terrestrial C sequestration (Bradford et al., 2016). There have been increasing efforts in taking into account microbial (enzymatic) dynamics and mineral association in soil organic C (SOC) models, such as CORPSE (Sulman et al., 2014), MIMICS (Wieder et al., 2014), MEND (Wang et al., 2014) and RESOM (Tang and Riley, 2014). Inclusion of these processes in SOC models has demonstrated possibilities to represent SOC responses to global warming (Sulman et al., 2018). Moreover, further inclusion
of the explicit vertical resolution of biogeochemical processes and transport allows for the reconciliation of the SOC depth and $^{14}C$ profile (Ahrens et al., 2015). Although these new microbial SOC models better describe C formation and turnover processes than the conventional models, they still lack full coupling with the N and P cycles.

The main challenge in coupling C and nutrient cycles in microbially explicit models is to account for the large stoichiometric imbalances between the microbial decomposers (i.e. soil microorganisms) and their resources (i.e. plant litter and SOM) (Xu
et al., 2013; Mooshammer et al., 2014). Soil microbial communities can adapt to these imbalances by adjusting their C:N:P ratios, typically through shifting community structure (e.g. fungal:bacterial ratios) (Rousk and Frey, 2015) or through eliminating excess elements by altering their use efficiencies (e.g. C use efficiency) (Manzoni et al., 2012). A well-known adaptive mechanism to these imbalances is the exudation of extracellular enzymes to release nutrients through hydrolysis (Olander and Vitousek, 2000; Allison and Vitousek, 2005) or enhanced SOM oxidation, known as the "rhizosphere priming effect" (Craine
et al., 2007). Recent evidence has also shown that soil P availability regulates phosphatase synthesis (Fujita et al., 2017) and influences SOM turnover (Lang et al., 2017). As the above-mentioned processes/phenomena are receiving more attentions, an increasing number of emerging microbially explicit models have started to tackle these challenges by accounting for the N cycle, enzymatic biosynthesis and rhizosphere priming (Abramoff et al., 2017; Sulman et al., 2017; Huang et al., 2018; Sulman et al., 2019) using certain novel approaches. For instance, the microbial adaptation concept has been applied to represent

the adaptation of enzyme allocation by microorganisms to maximise their growth through altering the preferential source of decomposition between plant litter and SOM, as demonstrated using the SEAM model (Wutzler et al., 2017).

Another emerging challenge of representing nutrient processes in microbially explicit models is the competition for nutrient uptake between plants and microbes (Dannenmann et al., 2016; Zhu et al., 2017). Regarding P, in particular, the soil mineral

surface adsorbs inorganic P to compete with plants and microbes (Bünemann et al., 2016; Spohn et al., 2018). The equilibrium chemistry approximation (ECA) approach has been proposed to simulate the competition of substrate uptake kinetics in complex networks where the uptake kinetics of one substrate affects the others (Tang and Riley, 2013). ECA has also been applied to resolve mineral nutrient sink (plant–microbe uptake or mineral adsorption) competitions in other modelling studies (Zhu et al., 2016, 2017, 2019).

In this study, we present the structure and basic features of a novel microbially explicit and vertically resolved SOM model that integrates with the N and P cycles—the Jena Soil Model (JSM). JSM combines the representations of the vertical structure, microbially explicit decomposition and stabilisation (Ahrens et al., 2015) with the microbial adaptation concept from the SEAM model (Wutzler et al., 2017) and the ECA approach (Tang and Riley, 2013). We tested alternative hypotheses regarding the competition among microbial, plant and mineral nutrient sinks (uptake or mineral sorption) and evaluated the effects of nutrient

availability on the preferential decomposition of either nutrient-poor or nutrient-rich organic matter using observed soil C, N and P profiles in a temperate beech forest stand. Additionally, we evaluated the model's sensitivity to parameterisation and associated uncertainty to help understand these effects.

## 2    Material and methods

### 2.1    Model description

JSM is a soil biogeochemical model built on the backbone of the vertically explicit C-only SOC model COMISSION (Ahrens et al., 2015). The COMISSION model was further developed from the conventional one by introducing a scalable maximum sorption capacity based on soil texture for dissolved organic C and microbial residues (Sect.S1.4) as well as temperature and moisture rate modifiers for microbe-mediated processes and sorption (Sect.S1). We will investigate in a separate study, how the maximum sorption capacity for mineral-associated organic C contributes to the observed patterns of SOC content and

radiocarbon age. A schematic overview of JSM is presented in Fig. 1, and the mathematical description of the processes is provided in Appendix A. The model is integrated into the QUINCY (Thum et al., 2019) TBM modelling framework and can either be applied as a stand-alone soil model or coupled to the representation of the vegetation and surface processes. In this study, we applied JSM as stand-alone model. JSM neither describes the energy and water processes at the atmosphere–soil interface or in the soil profile, nor simulates the production of litter. Model inputs (soil temperature, moisture and water fluxes

as well as plant litter data) were derived from the QUINCY model.

JSM describes the formation and turnover of SOM along a vertical soil profile, which is explicitly represented as exponentially increasing layer thickness with increasing soil depth (Fig. 1). The biogeochemical processes and pools of C, N and P are represented in each layer. Vertical transport of biogeochemical pools between the adjacent layers due to percolation and bio-

turbation is also modelled. To reflect the development of an organic layer, the model also includes an extra advective transport term which accounts for the upwards/downwards shift of the soil surface when the surface SOM accumulates/diminishes.

SOM is represented as pools of soluble, polymeric or woody litter as well as of dissolved organic matter (DOM), mineral-associated DOM, living microbial biomass, microbial residue (necromass) and mineral-associated microbial residue, each of which contains organic forms of C, N and P. The flows of organic N and P follow the pathways of C, with additional nutrient-specific processes, such as mineralisation and plant uptake, to link organic matter turnover with inorganic nutrient cycles. Microbial biomass is assumed to maintain a fixed stoichiometry in the model. It assimilates organic forms of C, N and P from DOM with fixed element use efficiencies and inorganic forms of N and P from soluble mineral pools. Microbes are assumed to aim to maximise their growth by maintaining high C use efficiency; however, when growth is limited by nutrients, microbes reduce their C use efficiency and increase nutrient mineralisation accordingly (See Sect.S1.5). The stoichiometry of all other SOM pools depends on the C:N:P ratios of influx and efflux, and these fluxes retain the stoichiometry of their source pools unless the formation processes involve respiration. In addition, when microbes decay, nutrients are preferentially recycled to the DOM pool due to the low C-to-nutrient ratio in the cyctoplasma, as proposed by Schimel and Weintraub (2003). The inorganic pools of N and P include soluble inorganic ammonium (referred to as $NH_4$), nitrate (refered to as $NO_3$), soluble inorganic phosphate (referred to as $PO_4$), as well as adsorbed $PO_4$, absorbed $PO_4$, occluded $PO_4$ and primary $PO_4$. The inorganic P cycle follows the QUINCY model (Thum et al., 2019) and accounts for microbial interactions. JSM explicitly traces $^{13}C$, $^{14}C$ and $^{15}N$ following Ahrens et al. (2015) and Thum et al. (2019).

Enzymes are not explicitly modelled in JSM, although these are described implicitly to regulate processes such as de-polymerisation and nutrient acquisition. For enzyme allocation within depolymerisation processes, we extended the microbial adaptation approach of the SEAM model (Wutzler et al., 2017) by including P dependence of enzyme allocation and the assumption of a steady state of enzyme production, leading to the prediction that the total enzyme level is always proportional to the microbial biomass. The fractions of enzymes allocated to different depolymerisation sources (litter and microbial residue) are dynamically modelled to maximise the release of the most limiting microbial elements. JSM tracks three potential fractions of enzyme allocation, which represent cases in which microbes only maximise depolymerisation release of C, N or P, respectively, and then updates the microbial enzyme allocation fraction by acclimating gradually to the potential fraction of most limiting element (See Sect.S1.5.2). For nutrient competition between plant, microbes and soil adsorption sites (only for phosphate), the potential rates are calculated on the basis of the respective richness and half-saturation level of enzymes and the impacts of other competitors, following the ECA approach (See Sect.S2.2).

The impacts of soil conditions on biogeochemical processes are also represented in JSM. The temperature response of different processes (e.g. microbial growth, decay, and nutrient uptake in Sect.S1.4) are represented by Arrhenius equation with different activation energies. Moisture responses are described by two rate modifiers—one representing the effects of oxygen limitation (e.g. litter turnover in Sect.S1.2) and the other representing the effects of diffusion limitation (e.g. depolymerisation in Sect.S1.3). JSM also considers the effects of SOM content to correct bulk density (Sect.S3), which in turn affects other processes such as organic matter (Eq.S7) and phosphate (Eq.S25) sorption.

## 2.2 Site description and data for model analysis

The Vessertal (VES) site is a mature beech (*Fagus sylvatica*) forest stand with an average tree age of >120 years, located in the central uplands of Germany (Thuringian Forest mountain range). The intermediate elevation is 810 m a.s.l, with a high annual precipitation of 1200 mm and a mean annual temperature of 5.5 °C (Lang et al., 2017). It is one of the Level II intensive monitoring plots in the Pan-European International Co-operative Program for the assessment and monitoring of air pollution effects on forests (ICP Forests). Since 2013, the VES site has also been one of the main study sites in the German Research Foundation (DFG) funded the priority programme 1685 'Ecosystem Nutrition: Forest Strategies for Limited Phosphorus Resources'.

Soil at the VES site is classified as Hyperdystric Skeletic Chromic Cambisol (WRB, 2015), with loamy topsoil and sandy loamy subsoil, overlain by a Moder organic layer. The current soil developed on trachyandesite, and the development started at the end of the last ice age, 10–12,000 years ago (Lang et al., 2017). The soil was sampled up to 1 m, with layer depths of 5–10 cm, for the measurements of total C, N and organic and inorganic P and basic physical properties such as bulk density and soil texture. Soil from the A horizon alone was extracted for the estimation of microbial C, N and P pools. Detailed sampling and measurement approaches are described in Lang et al. (2017).

The soil contains 19 $kg/m^2$ C, 1.1 $kg/m^2$ N and 464 $g/m^2$ P up to 1-m soil depth, including the forest floor (Lang et al., 2017). The soil C content decreases from 510 g C/kg soil in the forest floor to 126 g C/kg soil in the A horizon to 5.9 g C/kg soil at 1-m depth. The C:N ratio of SOM slightly decreases from 19.5 in the forest floor to 14.75 at 1-m depth, whereas the C:P ratio decreases more steeply from 348.7 in the forest floor to 46.6 at 1-m depth. The organic P fraction of total P also decreases from 67 % in the organic layer to 13 % at 1-m depth. The microbial C content decreases from >2000 µg C/g soil C in the forest floor (Zederer et al., 2017) to 764 µg C/g soil C in the top mineral soil (Bergkemper et al., 2016). The microbial biomass shows a C:N ratio of 13 and a very low C:P ratio of 10.3 (Lang et al., 2017).

## 2.3 Model protocol, model experiments and sensitivity analysis

**Model protocol**

The soil texture profiles for both QUINCY (for the generation of soil temperature, moisture and litterfall) and JSM simulations were obtained from observations at the VES site. The mineral-associated DOM and residue pools were initialised on the basis of Eq.S7 using the observed soil texture and mineral soil density, and assuming that the soil surface sorption sites are less occupied as soil depth increases. The vertical profile of the other SOM pools was initialised with a default C content for each pool in the first layer and assumed to decrease with soil depth in proportion to the fine root profile (Jackson et al., 1996), except in the woody litter, which is only initialised in the first layer. The initialisation C contents in the first layer for soluble litter, polymeric litter, woody litter, DOM, microbes and microbial residue were 291, 2914, 1000, 2.4, 73.2 and 203 $g/m^3$ C, respectively. The N and P contents of the SOM pools were initialised using the stoichiometry of different pools. For litter pools, we adapted the litter stoichiometry from the QUINCY model (Thum et al., 2019); for microbes and microbial residues, we used the measured microbial stoichiometry (Bergkemper et al., 2016) and for other SOM pools, we used the measured average

SOM stoichiometry of the 1-m soil profile (Lang et al., 2017). All SOC profiles were initialised with a pre-industrial $\Delta^{14}C$ values for all C pools, from which the $^{14}$C values were developed. The soil inorganic P pools were initialised using the soil P dataset from Yang et al. (2014a), corrected with the current total inorganic P from field measurements and extrapolated to the whole soil profile following the approach used in the QUINCY model (Thum et al., 2019). Organic matter material and mineral soil densities were solved using the Federer equation (Federer et al., 1993) with field data of the SOM content and bulk density.

We first ran the QUINCY model for 500 years to generate soil forcing and then simulated the VES site for 200 years using JSM, repeating 30 years of soil forcing (half-hourly soil temperature, soil moisture, vertical water fluxes and vertically resolved litterfall that includes $^{14}$C values) simulated by the QUINCY model for the VES site. To mimic the history of $^{14}$C input, we increased litter $^{14}$C content for the final 60 years before the end of the simulation, assuming that the $\Delta^{14}C$ in gross primary productivity in response to the observed $^{14}CO_2$ atmospheric pulse propagates directly into litterfall without any delay. We tested different simulation lengths (50, 200, 1000, 5000 and 10000 years) and observed that the simulated SOM profiles changed slowly after 200 years but the soil inorganic P pools changed gradually as the simulation time increased (Fig.2B). In the view of computational efficiency, we sought to compare the present-day soil profile observations with the simulated profiles for 200 years, which also best fit the date of soil inorganic P pool initialisation (1850, as indicated in Yang et al. (2014a)). All the other presented results (including sensitivity analysis) are also based on the 200-year simulations, and the results of long-term simulations (1000, 5000 and 10000 years) are specified with their simulation times.

**Model experiments**

To further test the effects of different model features, we implemented several model experiments, including a *SEAM-off* scenario in which the enzyme allocation to polymeric litter and microbial residue are both fixed to 50% (Eq.1b), and an *ECA-off* scenario in which the ECA-based plant and microbial uptakes of inorganic N & P and soil adsorption of phosphate were switched off and replaced by a demand-based microbial uptake of inorganic N & P that ignored P adsorption flux (Eq.1c). All model experiments used the same parameterisation from the calibrated model with full model features, which is denoted as the *Base Scenario* in this study.

The differences between *Base Scenario* and *SEAM-off* & *ECA-off* are listed below:

*Base Scenario* :

$$Enz_{frac}^{poly} \& \ Enz_{frac}^{res} \text{ calculated as Eq.S15}$$

$$U_{X,y}^* \text{ for microbes, plant and adsorption calculated as Eq.S23} \tag{1a}$$

*SEAM − off Scenario* :

$$Enz_{frac}^{poly} = Enz_{frac}^{res} = 0.5 \tag{1b}$$

*ECA − off Scenario* :

$$U_{X,plant}^* = f(T_{soil}, \Theta) v_{max,plant}^X C_{fine\_root}[X](K_{m1}^{upt} + \frac{1}{K_{m2}^{upt} + [X]})$$

$$U_{X,mic}^* = F_{mic,X}^{demand}$$

$$U_{P,adsorp}^* = 0 \tag{1c}$$

The plant uptake of inorganic N or P ($U_{X,plant}^*$) in the ECA-off scenario (Eq.1c) uses the equations and parameters from the QUINCY model (Thum et al., 2019).

**Calibration and model sensitivity**

We calibrated the *Base Scenario* in two main steps. In the first step, we matched the model results with the measured SOC profile, mainly by calibrating the depolymerisation, organic matter sorption and litter turnover processes; in the second step, we matched the model results with the measured soil organic P profiles by calibrating the microbial growth & decay, nutrient acquisition and soil inorganic P cycling. The two steps were not performed iteratively; however, during the second step, we revised the parameters from the first step as necessary. Other observed soil profiles, such as the soil organic N and the bulk

density, were used as additional criteria to select parameterisation, although not specifically used to calibrate the model. During the calibration processes, parameter values were gradually changed and the goodness of model fit was visually evaluated on the basis of observations.

To test the effects of different microbial stoichiometry, we ran a *Glob Mic Stoi* scenario in which the global average microbial stoichiometry (42:6:1, Xu et al., 2013) was used to parameterise the model instead of the observed microbial C:N:P ratio

(10.3:0.8:1, Lang et al., 2017). To further test the model responses to different initial conditions, we ran the model with different initial SOM contents (50%, 75%, 150%, and 200% of the default initial content) for 1000 years to ensure that the soil reached a more stable state.

We also tested the sensitivity of JSM to parameterisation using a hierarchical Latin hypercube design (LHS, Saltelli et al., 2000; Zaehle et al., 2005). We selected 28 parameters from calibration (Tab. S1) and varied each parameter between 80% and

120% of the base scenario values given in Tab. S2, which were obtained through LHS sampling from a uniform distribution to form a set of 1000 LHS samples and used in model sensitivity and uncertainty analyses presented in this paper. We evaluated the model output from these simulations in terms of main biogeochemical fluxes (e.g. respiration, net N & P mineralisation, microbial uptake of inorganic N & P, N & P losses and P biomineralisation) and main SOM pools (up to 1-m depth) (e.g.

total C, N and P in SOM; total soil inorganic P and microbial C, N and P). We measured parameter importance as the rank-transformed partial correlation coefficients (RPCCs) to account for potential non-linearities in the association between model parameters and output (Saltelli et al., 2000; Zaehle et al., 2005).

## 3 Results

 ### 3.1 Model stability and quasi-equilibrium state

We tested JSM with different initial SOM contents and different microbial stoichiometry. The simulated SOM profiles, including SOC; C:N and C:P ratios of SOM; microbial C, N and P contents and bulk density, did not respond strongly to changes in initial SOM contents (Fig.S2) but were notably affected by the assumed microbial stoichiometry (Fig.S1). We further examined the effects of simulation time on soil profile development (Fig.2 and Fig.S1). SOC in the topsoil (30 cm) reached a stable state (ca. 70 $kg\ C/m^3$) after approximately 150 years and the subsoil (30–100 cm) reached a stable state (ca. 30 $kg\ C/m^3$), after approximately 1000 years. The accumulation rate of SOM decreased with time, but the complete soil profile had not yet reached a steady state (Tab.1) because C continues to accumulate slowly, particularly in deeper soil layers (>1 m). Although the organic P dynamics follow the soil C dynamics, the inorganic P pools inevitably deplete in the long-term simulation (Fig.2) due to high uncertainties in initialisation and P cycling processes. Therefore in this study, we focussed on the stable state of topsoil (30 cm) at the end of the 200-year simulations and referred to it as a 'quasi-equilibrium state' since slow changes are still occurring, particularly in soil inorganic P pools and SOM in deeper soil layers (Fig.S3, Fig.S4 and S5).

### 3.2 Simulated SOM stocks and fluxes at the study site

We first compared the simulated profiles with the *in situ* observed ones (Fig.3). The modelled results agreed well with observed stock sizes and vertical patterns, indicating that the stocks [here we define the term 'stock' as the total amount of all (model) pools within a larger set] of C, N and P pools in SOM show smaller temporal variations than the microbial pools at the quasi-equilibrium state (Fig.3a to 3c) due to strong seasonal variations in microbial biomass. We also found a greater variation in the simulated organic P-to-inorganic P (Po-to-Pi) ratio (Fig.3d) than for the individual organic and inorganic P stocks (data not shown), inferring that the seasonal dynamics of microbes also impose a seasonal pattern of P immobilisation (from Pi to Po) and mineralisation (from Po to Pi).

The distribution and radiocarbon profile of total organic matter in the simulations varied across soil depths (Fig.4). The first layer (0 cm, O–A horizon) is dominated by the plant litter and microbial component (living/dead microbes), and while the microbial component decreases strongly from ca. 40% at 0 cm to almost zero at 50 cm soil depth, the litter component still constitutes ca. 10% of the total SOC at 1-m soil depth. The mineral-associated C (MOC) component switches from a minor component in the O–A horizon (ca. 20%) to the dominant component (ca. 90% at 1 m) in deeper soil layers.

The simulated radiocarbon ($\Delta^{14}C$) profile qualitatively agreed with observed one (Fig.S1e); the $\Delta^{14}C$ content increased within the O horizon and started decreasing with increasing soil depth from mineral soil, i.e. the A horizon. This pattern

indicates that the 'bomb pulse' $\Delta^{14}C$ signal significantly affects the apparent $^{14}C$ age in the organic layer and its impact decreases with soil depth due to the slow turnover in deeper soil. Our simulations further indicated that such a vertical pattern is caused by MOC and microbial components, while the litter component stays modern throughout the profile (Fig.4). Increases in litter $^{14}C$ with depth suggest that more bomb-derived SOC is still found in subsoils due to slower litter turnover, while it is already replaced by more recent, $^{14}C$-poorer SOC in the topsoil. Although the base scenario did not reproduce the measured radiocarbon profile, albeit only its vertical pattern, a much better fit with the measured radiocarbon profile and increase in soil $^{14}C$ age, driven by MOC, were indeed observed as simulation time increased (Fig.S1e and Fig.S4).

The simulated biogeochemical fluxes show strong seasonal and vertical patterns (Fig.5 and Fig.6), in which the flux rates in summer and in the topsoil are generally higher than those in winter and in the subsoil, respectively. Meanwhile, microbial inorganic N uptake shows a different seasonal pattern, with the lowest rates observed in August and September (Fig.6c). In fact, this pattern is supported by the seasonal pattern of net N mineralisation flux, in which the peak is observed in August and September (Fig.5b). This result indicates that organic N in DOM is the most abundant for microbial growth during August and September, leading to a large reduction in the microbial inorganic N uptake and increase in net N mineralisation. In contrast, organic P content in DOM is the lowest during August and September, leading to net P immobilisation and microbial inorganic P uptake elevation (Fig.5d and Fig.6a). While the vertical pattern of plant N uptake parallels root distribution (Jackson et al., 1996), plant P uptake is lower in the organic layer than in the topsoil due to strong competition from microbes in the organic layer (Fig.6 and Fig.8).

The sources and sinks of soluble inorganic N and P also show very different patterns (Fig.7). The main source and sink for inorganic N in solution are gross mineralisation and plant uptake of $NH_4$, respectively; whereas for P, microbial uptake is the main sink and biomineralisation is a larger source than gross mineralisation in each scenario.

### 3.3 Model features: nutrient acquisition competition and enzyme allocation

In general, the SEAM-off scenario did not differ much from the base scenario in terms of the main soil stocks and biogeochemical fluxes (Fig.3 and Fig.5); however, the ECA-off scenario produced a lower microbial biomass and Po-to-Pi ratio in the organic layer and topsoil. Total SOC may not be influenced in either scenario, although its composition and radiocarbon profile were both altered (Fig.4).

We presented the uptake of inorganic $PO_4$ and competition between phosphate adsorption, uptake of inorganic P at three different depths (Fig.8) and seasonal and vertical uptakes of inorganic N & P for both microbes and plants (Fig.6). The simulations showed that microbes outcompeted roots for inorganic P uptake in JSM at all depths. However, the relative competitiveness of roots for phosphate uptake increased with increasing soil depth because the plant P uptake rate decreases less strongly than the microbial P uptake with increasing soil depth. In contrast, the phosphate adsorption rate increased strongly with increasing soil depth and outcompeted biological processes (plant and microbial uptake) in deeper soil layers. The relative competitiveness of phosphate adsorption against microbial and plant uptake also strongly decreased in summer in the topsoil due to high biological activity in warm months (Fig.8B). With respect to competition for inorganic N, plants outcompeted microbes along the entire soil profile and throughout the year, particularly in summer when microbes assimilate N mainly from DOM (Fig.6c and d).

Turning off the model's feature for nutrient acquisition competition, i.e. ECA-off scenario, led to a notably lower microbial biomass and Po-to-Pi ratio in the topsoil (Fig.3). This is caused by concurrent changes in microbes and plant inorganic P uptake, particularly in the topsoil layer where plants take up more inorganic P than in the base scenario (Fig.6) due to reduced competition with microbes. Moreover, there were differences in spatial and temporal variations in uptake and mineralisation fluxes between the ECA-off scenario and the other two scenarios. For instance, the seasonal variation in fluxes was notably lower in the ECA-off scenario. Decrease in P flux rate with soil depth may be weaker in the ECA-off scenario, but decrease in net N mineralisation with soil depth is marginally stronger (Fig.5). This difference is because that geophysical processes, such as adsorption and absorption, play more crucial roles in the soil P cycle than in the N cycle and that these show rather different seasonal and vertical patterns from the biochemical processes, such as mineralisation.

The modelled enzyme allocation for depolymerisation is presented in Fig.9. We compared the enzyme allocation curve of polymeric litter ($Enz_{frac}^{poly}$ in Eq.S17) with three potential allocation curves ($\alpha_{poly}^{X}$ where $X$ stands for C, N, and P, in Eq.S15), which represent cases in which microbes only maximise C, N or P release from depolymerisation. All modelled fractions of enzyme allocation to polymeric litter were well below 50% for the whole soil profile, indicating that polymeric litter is less preferred than microbial residues for depolymerisation in the soil, particularly in very deep soil layers where no roots are present and microbes would thus only produce enzyme to depolymerise microbial residues because the content of residue is much higher than that of polymeric litter. The simulated curve of allocation overlaps with the curve of potential allocation to maximise P release, indicating that depolymerisation is solely driven by P demand. This explains why microbial residues are preferred over polymeric litter since the C:P ratio of microbial residues is much lower than that of polymeric litter (data not shown). Despite rather different enzyme allocation fractions shown in Fig.9, majority of the modelled stocks and fluxes were not significantly influenced when enzyme allocation was turned off (Fig.3 and 5). More profound differences were observed in the composition and radiocarbon profile of SOC; there was less litter and more SOC in the SEAM-off scenario than in the base scenario, resulting a systematic difference in the radiocarbon profiles between the two scenarios (Fig.4).

### 3.4 Model sensitivity and uncertainties

The interquartile range of outputs (Fig.10) from model sensitivity analysis revealed that all outputs were well centred around the results of the parameterisation of the base scenario (Tab.S2), except microbial inorganic N uptake and N losses. In general, the soil stocks were more stable than the microbial pools and biogeochemical fluxes. N mineralisation was surprisingly insensitive while microbial inorganic N uptake was very sensitive to parameterisation. N mineralisation in JSM was mainly driven by the C:N ratio of DOM, which remains rather stable due to the similar C:N ratios of plant litter, microbes, and microbial residues. The very sensitive response of microbial inorganic N uptake was attributed to the high affinity (low $K_{m,mic}$ value) N uptake transporters of microbes (Kuzyakov and Xu, 2013) and their sensitivity to changes in $NH_4$ concentration. The RPCC of parameters with outputs (Tab.2) also demonstrated that the C and N contents of SOM as well as the C and N fluxes were more sensitive to changes in C processes, such as depolymerisation, organic matter sorption and litter partitioning, while the microbial dynamics and the P fluxes were more sensitive to changes in microbial and nutrient processes, such as maximum biomineralisation rate ($v_{max,biomin}$) and P recycling during microbial decay ($\eta_{res \to dom}^{P}$). Overall, most of the

selected outputs were strongly influenced by microbial stoichiometry. The five most influencing parameters in JSM were microbial C:N ratio ($\chi_{mic}^{C:N}$), microbial N:P ratio ($\chi_{mic}^{N:P}$), microbial mortality rate ($\tau_{mic}$), soluble litter C fraction transformed into DOM ($\eta_{C,sol \to dom}$), and P fraction recycled from $res$ to $dom$ during microbial decay ($\eta_{res \to dom}^{P}$).

## 4 Discussion

### 4.1 Features of nutrient cycling

**Soil stoichiometry**

Following calibration, JSM could reproduce the main soil stocks of C, N and P; microbial biomass and soil bulk density as well as their vertical patterns along the soil profile in a beech forest stand in Germany. The observed SOM C:N ratio (19.5) and C:P ratio (348) in the first model layer—the O–A horizon—fit well within the ranges of reported soil stoichiometry of temperate broadleaf forests (Xu et al., 2013), and there was a much stronger decreasing trend of C:P ratio than C:N ratio with increase in soil depth, indicating that organic P in SOM is 'decoupled' from the C and N cycles (Yang and Post, 2011; Tipping et al., 2016).

This decoupling of the soil P cycle is represented by biomineralisation in TBMs; however, the vertical decoupling of C:N:P stoichiometry is poorly reproduced (Fig.S6) even when microbial biomass is explicitly represented (Yu et al., 2018). Our study indicated that the decrease in C:N ratio is mainly due to a shift in SOC composition with soil depth (Fig.4), whereby fraction of the nutrient-poor litter component decreases and that the nutrient-rich MOC component increases. Slight overestimation of the modelled soil C:N ratio in the first layer (Fig.3) is probably due to the higher C:N ratio (52) of leaf litter inputs than the observed one (41.7).

However, with respect to the decrease in C:P ratio, the model simulations indicated that the change in microbial nutrient recycling scheme with depth might be associated with shift in the SOC component. To account for different stoichiometry of cell walls and plasma of microbes in JSM, we introduced the microbial nutrient recycling parameter ($\eta_{res \to dom}^{X}$, X for N or P) that partitions microbial residues with lower C:N:P ratio according to P stoichiometry, that is, a higher nutrient content is allocated to DOM, while the residual pool receives the remaining part with a lower nutrient content. Since JSM currently does not distinguish among microbial guilds, the microbial nutrient recycling parameters also mimic different stoichiometry of microbial guilds, such as bacteria and fungi. The model only adequately reproduced the vertical SOM C:P ratio profile when the microbial P recycling parameter decrease with increasing depth, resulting in a decreased C:P ratio with increasing soil depth. Such a shift in the microbial P recycling parameters indicates changes in microbial communities from nutrient-poor fungi-dominated to a nutrient-rich bacteria-dominated one with increasing depth, which has also been evidenced by Rousk and Frey (2015). Our model suggests that this community shift mainly regulates decrease in SOM C:P ratio at the study site.

**N cycle vs. P cycle**

JSM had already reached a quasi-equilibrium state at the end of the 200-year simulation, when the respiration of C and plant uptake of N and P were very close to the C, N, and P from litterfall and SOM accumulates slowly in the soil (Tab.1, Fig.2). As the simulation time increased, the C and N cycles approached true equilibrium but the P cycle did not (Tab.1); this could be

due to the lack of vegetation feedback, or the constantly increasing occluded P pool and decreasing primary P pool that do not allow to reach true equilibrium in JSM. Similar trend have been observed with all TBMs because they employ the structure of inorganic P cycle described in Wang et al. (2010) is used. This leads to a boundary issue, particularly in long-term simulations, and warrants further investigation, particularly for the development of soil profiles.

5 In JSM, plant nutrient uptake is driven by root biomass (prescribed by the QUINCY outputs) and its uptake capacity (as reported in (Kuzyakov and Xu, 2013; Kavka and Polle, 2016)). Plant uptake is further influenced by microbial (P adsorption) competition, but it is not regulated by plant demand due to the absence of vegetation processes. The simulated plant N and P uptakes at the quasi-equilibrium state were very close to the N and P inputs from the litterfall (Tab.1), indicating that realistic root biomass and uptake capacity enable the simulation of nutrient uptake for plant. This finding supports the recent change
10 in plant uptake simulation in TBMs from plant demand driven (Yang et al., 2014b) to trait (root biomass, uptake capacity, and inorganic nutrient pool) driven (Zaehle and Friend, 2010; Goll et al., 2017; Thum et al., 2019), which strengthens the interactions between soil nutrient availability and plant growth.

 The simulated microbial uptake of inorganic P (238.0 $\mathrm{kgP/ha/yr}$) was much higher than the plant inorganic P uptake (8.5 $\mathrm{kgP/ha/yr}$) and microbial inorganic N uptake (Fig.7). This difference was strongly driven by the difference between
15 litterfall and microbial stoichiometry. In JSM, nutrient assimilation for microbial growth occurs at two steps. In the first one, a certain fraction of N & P ($mic_{nue}$ and $mic_{pue}$) from microbial DOM uptake is assimilated directly by microbes; in the second step, dissolved inorganic N & P are further taken up by microbes through microbial inorganic N & P uptake to fulfil their stoichiometry. In the base scenario, we used the measured microbial C:N:P ratio at the study site (10.3:0.8:1), which largely differs from the litterfall C:N:P ratio (800:14.8:1), particularly in terms of the P content. Therefore, although the demand for N
20 and P for microbial growth does not differ much, the assimilation of dissolved organic N is much higher than that of dissolved organic P, resulting in a much higher demand for microbial P uptake than for N uptake from the inorganic pool and very different seasonal patterns of microbial inorganic N and P uptakes (Fig.6). This is well demonstrated in Fig.5 and Fig.7; net mineralisation, calculated by subtracting microbial inorganic nutrient uptake from gross mineralisation, is always positive for N and mostly negative for P, particularly in the warm season when microbial biomass is high. While majority of the mineralised
25 N is taken up by plants, only a minor fraction of mineralised P is taken up by them, and most of it, together with the additional biomineralised P, is taken up by microbes in the form of dissolved inorganic P. This pattern implies that the mobilisation of soil N is driven by plant demand and that of soil P is driven by microbial demand.

**Microbial stoichiometry**

 Since the microbial C:N:P ratio we used (10.3:0.8:1, Lang et al., 2017) was very different from the global average value
30 (42:6:1, Xu et al., 2013), additional modelling experiments were conducted with the global microbial stoichiometry to examine the effects of this ratio (Fig.S1 and Fig.S3–5). The SOC and microbial C profiles did not differ significantly in the new scenarios, although the N & P stocks and fluxes were greatly influenced. As a direct consequence of a change in microbial stoichiometry change, the resultant SOM C:N and C:P ratios became lower and higher, respectively, than values in the base scenario. Moreover, the total demand for microbial N was much higher and the demand for microbial P was much lower than
35 that in the base scenario, leading to a higher microbial inorganic N uptake and lower microbial inorganic P uptake, which

in turn alter the plant–microbe competition for inorganic N & P as well as the vertical and seasonal patterns of plant and microbial uptake of inorganic nutrients. Although the microbial P demand was lower in the scenario with the global microbial stoichiometry than in the base scenario, it still drove the soil P mobilisation. However, N mobilisation in the new scenario was no longer exclusively plant driven and became both microbe and plant driven. This indicates that the microbial stoichiometry is a key factor for soil nutrient processes and plant–soil interactions in JSM.

In JSM, the choice of nutrient mineralisation–immobilisation pathways (Manzoni and Porporato, 2009) during microbial DOM uptake, i.e. the microbial nutrient use efficiencies in Eq.S13, did not greatly change the total microbial nutrient assimilation but significantly impacted the partitioning between organic (microbial DOM uptake) and inorganic (microbial inorganic nutrient uptake) nutrient assimilation (Tab.2). This partitioning greatly alters the isotopic signals of soil pools and is essential to understand soil nutrient cycling and thus to unravel soil effects based on vegetation signals (Craine et al., 2018)— something which is not possible with the current TBMs due to poorly defined and parameterised microbial nutrient use efficiencies (Manzoni and Porporato, 2009). It is possible to use JSM to predict realistic microbial nutrient use efficiencies with constraints of tracer experiments by labelling different forms of dissolved nutrients. However, future detailed investigation is needed due to complications arising from other involved processes such as adsorption/desorption and nitrification/denitrification.

## 4.2 Key features and model limitations

We applied the ECA approach described by Tang and Riley (2013) to simulate inorganic nutrient competition. In general, our model simulations indicated that microbes take up more inorganic P than plants, which supports the findings of $^{33}P$ tracer experiments at two other beech forests in Germany (Spohn et al., 2018). However, our study showed that plants take up more inorganic N than microbes (Fig.7A and Fig.S1). This pattern seems to disagree with the findings of field studies of $^{15}N$ addition (e.g. Bloor et al., 2009; Dannenmann et al., 2016) and a modelling study using the same approach to simulate competition (Zhu et al., 2017). The reason for this disagreement is that in JSM, we assumed high microbial N use efficiency from DOM and majority of microbial N assimilation was actually fulfilled by DOM uptake. Therefore, plants take up more inorganic N than microbes. However, in $^{15}N$ tracer experiments and a model study by Zhu et al. (2017), there was no distinction between assimilation from organic and inorganic sources; thus, microbes outcompete plants in the sense that the total N assimilated by microbes exceeds the total N taken up by plant roots, which was also true in our study. Another uncertainty related to the plant–microbe competition for inorganic N is the microbial stoichiometry we used in parameterisation. As discussed in the previous section, a change in microbial stoichiometry from the observed field value to the global average value resulted in a switch from microbes outcompeting plants for inorganic N to the opposite trend. Additionally, the choice of microbial nutrient use efficiencies not only affected the microbial demand for inorganic nutrients and the concentrations of inorganic N & P, thereby influencing the potential uptake rates of microbes and roots.

We extended the enzyme allocation approach of the SEAM model (Wutzler et al., 2017) by including P dependence and vertical explicitness and by assuming a steady state of enzyme production. Due to the very small microbial C:P ratio used in model parameterisation, our results indicated that depolymerisation is solely driven by P demand; thus, microbial residues are the preferred substrate because they have a much lower C:P ratio than polymeric litter. This is also supported by the massive

P biomineralisation flux (Fig.7) independent of depolymerisation and gross mineralisation, indicating that microbial growth is strongly P limited. Even in the scenario using the global microbial stoichiometry, depolymerisation was still solely P driven, and P biomineralisation fulfilled over half of the microbial P demand (Fig.S5). This result is partly supported by the global enzymatic activity data in which global ratios of specific C, N and P acquisition activities converged on 1:1:1 (Sinsabaugh

et al., 2008), while the global microbial stoichiometry was much higher, indicating that relatively more resources are allocated to acquire P than to acquire N and C. This result actually reveals a caveat in the current implementation of enzyme allocation in JSM that the main process via which organic P is hydrolysed, biomineralisation and the mobilisation of sorbed inorganic P due to root exudation are not included in the enzyme allocation calculation. It also explains the very small difference between the base scenario and the SEAM-off scenario.

JSM demonstrated a capacity to reproduce the vertical patterns of soil stocks (Fig.3) and to satisfactorily produce both vertical and seasonal patterns of biogeochemical fluxes (Fig.5 and 6). While the seasonal patterns are primarily driven by the temperature response of the represented processes, the vertical patterns are shaped by the combined effects of biochemical and geophysical factors represented in the model. As seen in Fig.3 and Fig.4, although total SOC decreased with soil depth, the microbial, litter and MOC components showed very different patterns. Following the COMISSION model(Ahrens et al., In

prep.), we constrained the capacity of organo-mineral association with silt and clay contents and soil bulk density in JSM. In the organic layer and topsoil, the continuous litter input sustains a large microbial biomass and microbial residue pool; however, due to the very low bulk density and relatively low silt & clay contents, sorption is weak and MOC content is very low. As soil depth increases, bulk density and silt & clay contents increase such that microbial residues and DOM stabilised to a greater extent. This hinders microbial DOM assimilation and nutrient immobilisation, leading to a strong decline in microbial biomass

and an increase in MOC. As a consequence of the decreasing microbial biomass and litter inputs, much less microbial residue and DOM are available for sorption to the mineral soil, which explains the observed decrease in total SOC in deep soil layers.

Nonetheless, certain caveats of this study and JSM should be discussed. A main challenge is the different simulation times for different purposes. Our results indicated that in the upmost 30 cm of soil, SOM content stabilises after 150 years while in the upmost 1 m SOM stabilises after 1000 years of simulation (Fig.2), regardless of the initial SOM content (Fig.S2). However,

with respect to the radiocarbon profile, as indicated by Ahrens et al. (2015), a very long simulation time (13500 years) was required to match both the measured $\Delta^{14}C$ and SOC profiles at a nearby Norway spruce forest site. In our study, a 10000-year simulation time was still not sufficient to match the measured $\Delta^{14}C$ profile, indicating that an even longer simulation time is required. Although JSM is very stable in the long term in term of SOM development and storage, long-term simulation of soil P balance as a result of continuous weathering and occlusion remains a significant challenge (Fig.2, Tab.1). Such a long

simulation time is unrealistic for the P cycle due to the unknown conditions of the initial soil P pools and the un-equilibrated soil inorganic P cycling processes (Yang et al., 2014b). Although we used a much shorter simulation length in this study, noticeable uncertainties remain due to inorganic P cycling parameters (Tab.2). Additionally, the long simulation time required to match the radiocarbon profiles is also problematic for future coupling to TBMs because these models typically examine centennial time scales. A possible solution is to spin-up radiocarbon (>10000 years) independent of the plant–soil spin-up (1000 years),

although this approach needs to be properly tested in the future.

Another caveat involves the model's representation of microbial adaptation schemes. In JSM, we describe enzyme allocation, which is one of the schemes of microbial adaptation proposed by Mooshammer et al. (2014); however, as discussed above, enzyme allocation to phosphatases might be essential and might thus need to be included. Additionally, we found out that another adaptation scheme, the microbial community shift between fungi and bacteria, is crucial for reproducing the vertical pattern of soil stoichiometry. Although we mimicked such a shift in this study by calibration and parameterisation, a more mechanistic representation is necessary in the future for representing the acclimation of microbial functional properties to climate and environmental changes.

Concerning the model's description of N dynamics, in the current version, N processes such as nitrification/denitrification and abiotic ammonium adsorption are not yet implemented. Although the simplified N dynamics will probably not alter the main findings of this study, it is important to investigate these in the future since plants often have a preference for ammonium uptake (Masclaux-Daubresse et al., 2010). Finally, given the good quality of the input data, JSM could adequately reproduce the soil stocks and flux rates at the selected study site; however, its capacity to extrapolate to other climate and soil conditions needs to be further investigated in the future.

JSM is highly non-linear and sensitive to the parameters controlling microbial growth and decay (Tab.2). The C and N stocks in SOM as well as respiration and net N mineralisation are highly sensitive to the parameter changes of depolymerisation and organo-mineral association, whereas the organic/inorganic P stocks and P mineralisation are highly sensitive to the microbial processes. These trends support, and also explain, the finding of Yang and Post (2011) and Tipping et al. (2016) that the P cycle is decoupled from the C and N cycles in the soil. A more in-depth explanation for this difference, based on our results, is that the gross mineralisation associated with microbial DOM uptake can supply microbes and plants with sufficient N but not P; thus, a large amount of P needs to be mobilised, particularly from SOM as well as from mineral pools, to sustain microbial growth. Therefore, the microbial pools and soil P stocks/fluxes are highly sensitive to microbial processes.

## 5 Summary and future directions

We presented the mathematical formulation for a new SOC model—JSM—which is an extension of the vertically explicit, microbial-based, and organo-mineral association-enabled SOC model, COMISSION, developed by introducing the N and P processes via novel approaches such as optimised enzyme allocation, nutrient acquisition competition, and process acclimation. The model was evaluated with the observed C, N and P stocks of SOM; soil inorganic P stock; microbial C, N and P contents and soil bulk density in the topmost 1-m soil in a beech forest stand in Germany. JSM captured the extents and vertical patterns of these observations. We further presented the main features of nutrient cycling under the new model structure and the sensitivities of model outputs to parameter changes; both indicated that the P cycle is largely decoupled from the C and N cycles and shows very close associations with microbial dynamics. Evaluation of model experiments underscores the need for improved representation of microbial dynamics in JSM, particularly their interactions with the P cycle.

To better represent microbial dynamics, we would need detailed and advanced understanding of microbial processes from experiments for implementation and testing in the model. For example, how will microbial C use efficiency change in response

to changes in C sources (e.g. DOM or litter addition) and nutrient availability (e.g. N & P addition)? How starkly does the microbial community adjust its stoichiometry, change its element use efficiency or alter extracellular enzyme synthesis under dynamic external conditions?

Next steps for evaluation of JSM are to investigate the effects of P cycling on microbial dynamics and SOM cycling in greater detail by subjecting it to other beech forest sites in Germany along a soil P availability gradient and to evaluate if the contrasting P cycling patterns proposed by Lang et al. (2017)—'acquiring system' and 'recycling system'—can be reproduced. Such a model evaluation is expected to identify the key/missing processes of the model to reproduce the contrasting P cycling schemes and to assess their effects on the SOM turnover/stability.

JSM was developed under the framework of the new biosphere model QUINCY, and the future plan is to apply this model coupled with the vegetation component of the QUINCY model described by Thum et al. (2019), which will offer an alternative to better represent the interactions between root growth/activity and SOM turnover and stabilisation in TBMs.

*Code availability.* JSM is developed using the framework of the QUINCY model and is licensed under GNU GPL version 3. The scientific code of JSM requires software from the MPI-ESM environment, which is subject to the MPI-M-Software-License-Agreement in its most recent form (http://www.mpimet.mpg.de/en/science/models/license). The source code is available online (https://git.bgc-jena.mpg.de/quincy/quincy-model-release, branch "jsm/release01"; doi:10.17871/quincy-model-2019), but access is restricted to registered users. Readers interested in running the model should request a username and password from the corresponding authors or via the git-repository. Model users are strongly encouraged to follow the fair-use policy stated at https://www.bgc-jena.mpg.de/bgi/index.php/Projects/QUINCYModel.

*Author contributions.* SZ and MS conceived the model. LY and SZ developed the model. MS measured the $^{14}C$ data. All authors contributed to the interpretation of the results and writing of the manuscript.

*Acknowledgements.* This work was supported by the framework of Priority Program SPP 1685 "Ecosystem Nutrition: Forest Strategies for Limited Phosphorus Resources" of the German Research Foundation (DFG), grant No. ZA 763/2-1 and grant No. SCHR 1181/3-1. We are grateful to Jan Engel for technical assistance in developing the code, and to Marleen Pallandt for improving the quality of the manuscript.

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

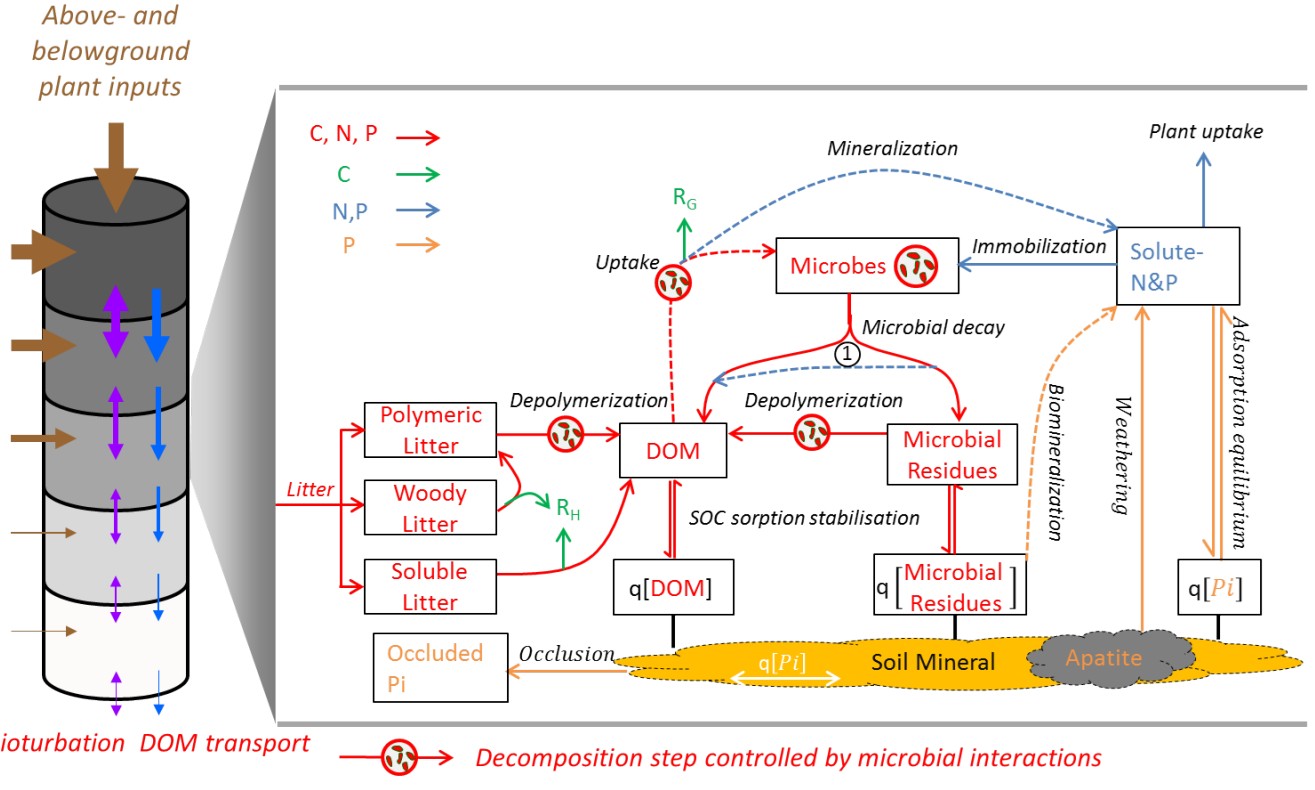

**Figure 1.** Theoretical representation of Jena Soil Model (JSM) structure. The vertical soil profile (9.5 m) is split into 15 soil layers; aboveground litter is added on top of the soil profile; root litter enters into each soil layer according to the root distribution. Bioturbation and DOM transport translocate SOM between soil layers. In each soil layer, boxes refer to pools and lines refer to processes, in which red lines: biogeochemical fluxes of C, N and P; green lines: respiration fluxes, $R_H$ for heterotrophic respiration and $R_G$ for microbial growth respiration; blue lines: fluxes of N and P; orange lines: fluxes of only P; dashed lines: biogeochemical processes that involves stoichiometry change between the sourcing and sinking pools. ①: microbial nutrient recycle from residue to DOM during decay; $q[X]$: mineral-associated form (adsorbed to soil mineral surface or absorbed into soil mineral matrix) of X, which can be DOM, microbial residues or inorganic phosphate (Pi).

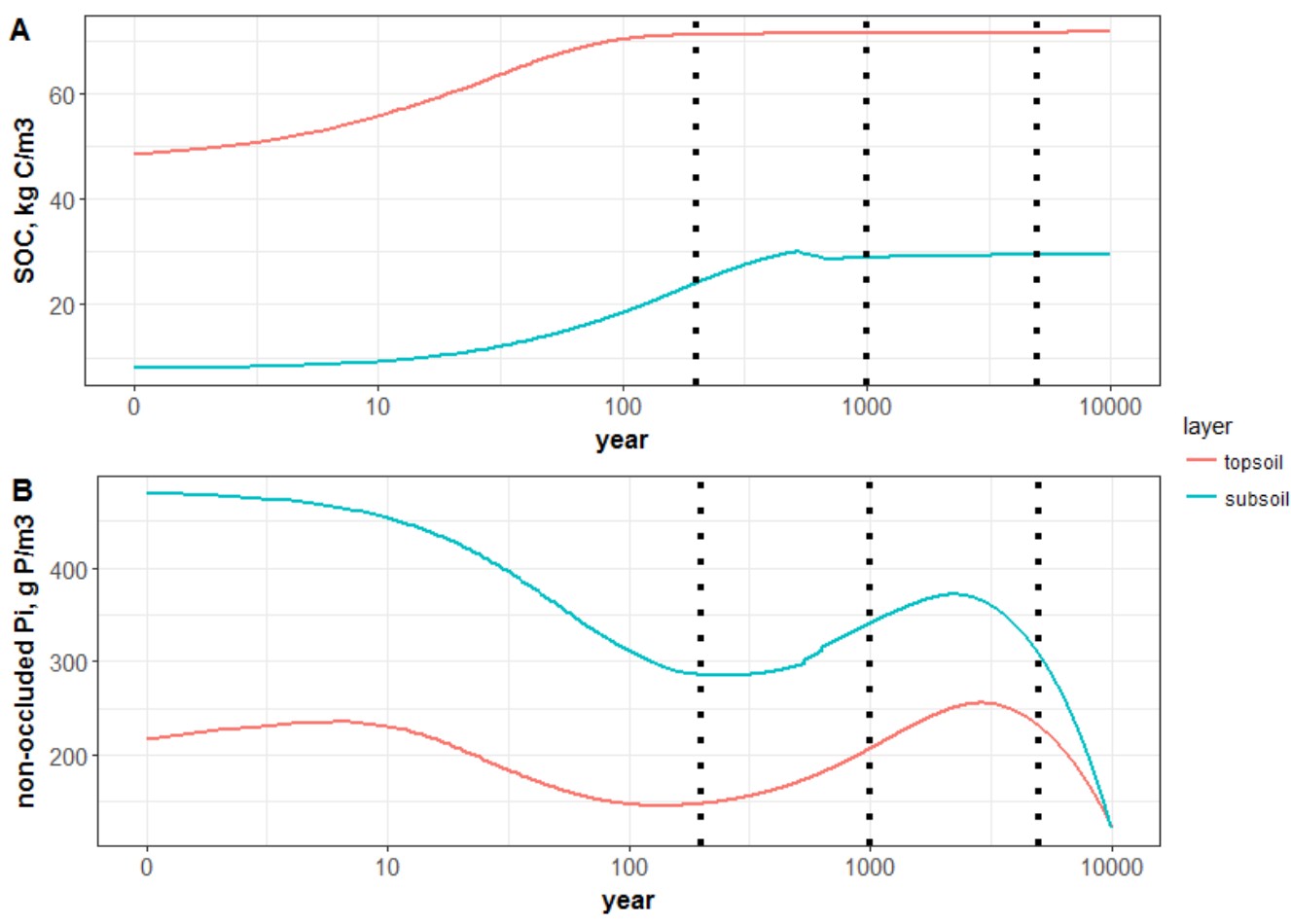

**Figure 2.** Simulated dynamics of (A) SOC and (B) non-occluded inorganic P contents in topsoil (30 cm) and subsoil (30–100 cm) for 10000 years. The three vertical dashed lines represent 200, 1000 and 5000 years, respectively.

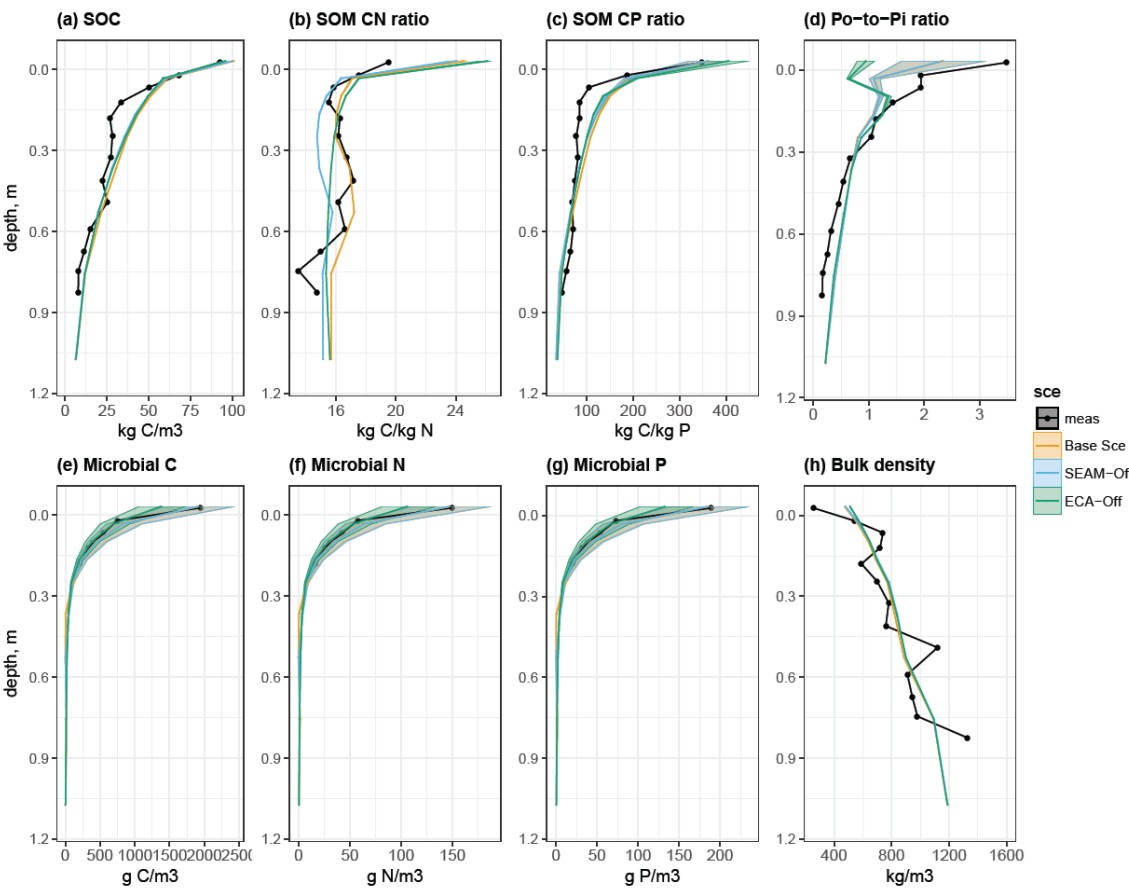

**Figure 3.** Simulated and observed (a) SOC content, (b) C:N ratio in SOM, (c) C:P ratio in SOM, (d) organic P-to-inorganic P ratio in soil, (e) microbial C content, (f) microbial N content, (g) microbial P content and (h) soil bulk density at the study site up to 1-m soil depth. Black lines and dots: observations; Coloured lines and shades: simulated mean values and ranges of standard deviation by different model experiments in Sect.2.3. Microbial C, N and P values were only measured in the top 30 cm of soil. Simulated means and standard deviations were calculated using data from the last 10 years of model experiments.

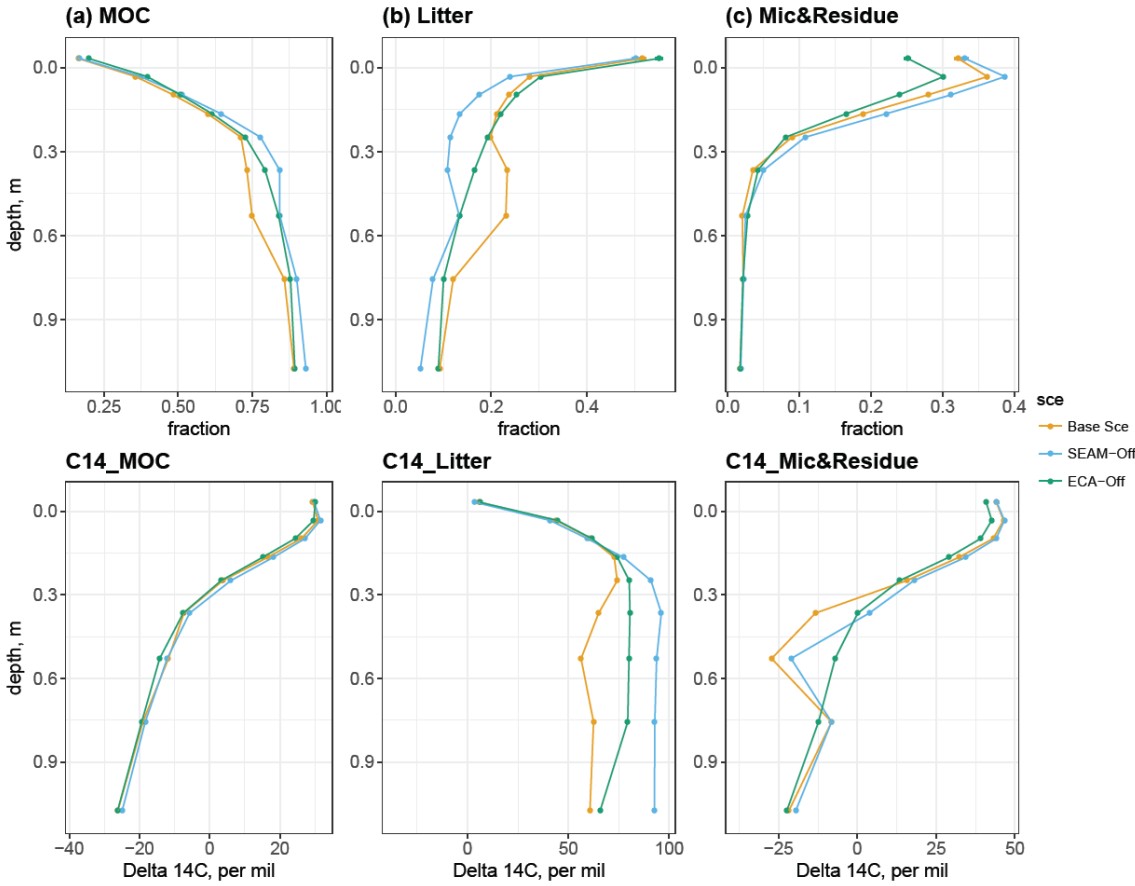

**Figure 4.** Simulated SOC fractions (upper panels) and their respective radiocarbon profiles (bottom panels) at 1-m soil depth. Column (a): mineral-associated C (MOC), including adsorbed DOM and adsorbed microbial residue; Column (b): litter, including woody, polymeric and soluble litter; Column (c): live and dead microbes. Data points are derived using data from the last 10 years of the model experiments. All model experiments used 200-year simulations and were not validated against the measured $\Delta^{14}C$.

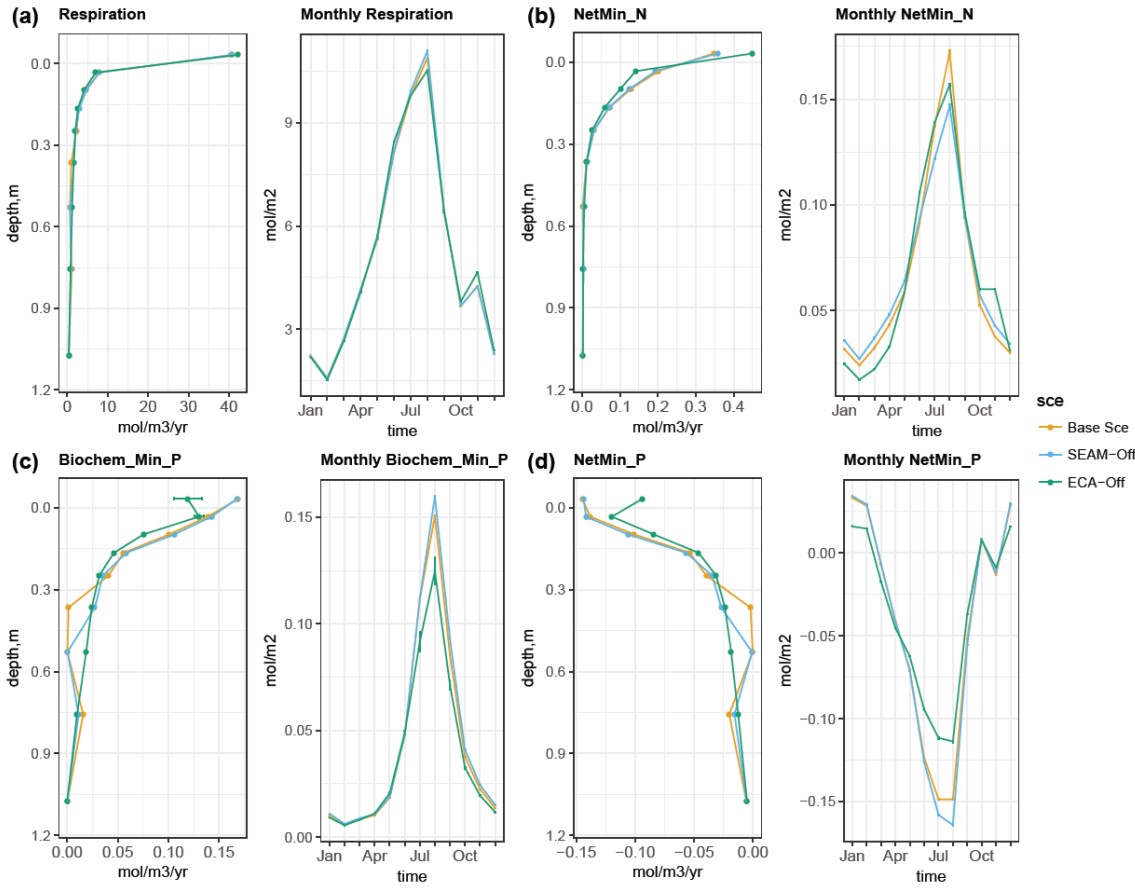

**Figure 5.** Simulated seasonal and vertical distribution of (a) respiration, (b) net N mineralisation, (c) biochemical P mineralisation and (d) net P mineralisation at the study site at 1-m soil depth. Points represent the mean values and error bars represent the standard deviations, both calculated using data from the last 10 years of model experiments.

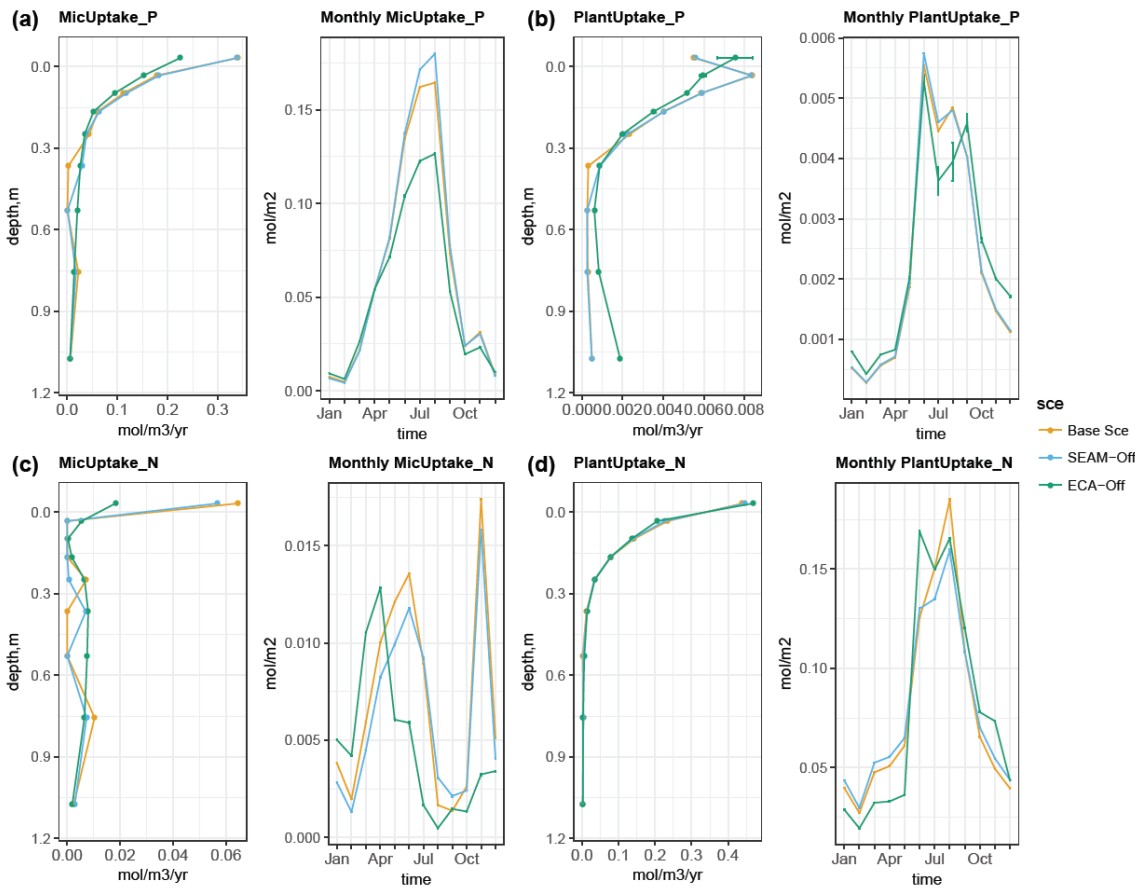

**Figure 6.** Simulated seasonal and vertical distribution of (a) microbial inorganic P uptake, (b) plant P uptake, (c) microbial inorganic N uptake, and (d) plant N uptake at the study site at 1-m soil depth. Points represent the mean values and error bars represent the standard deviations, both calculated using data from the last 10 years of model experiments.

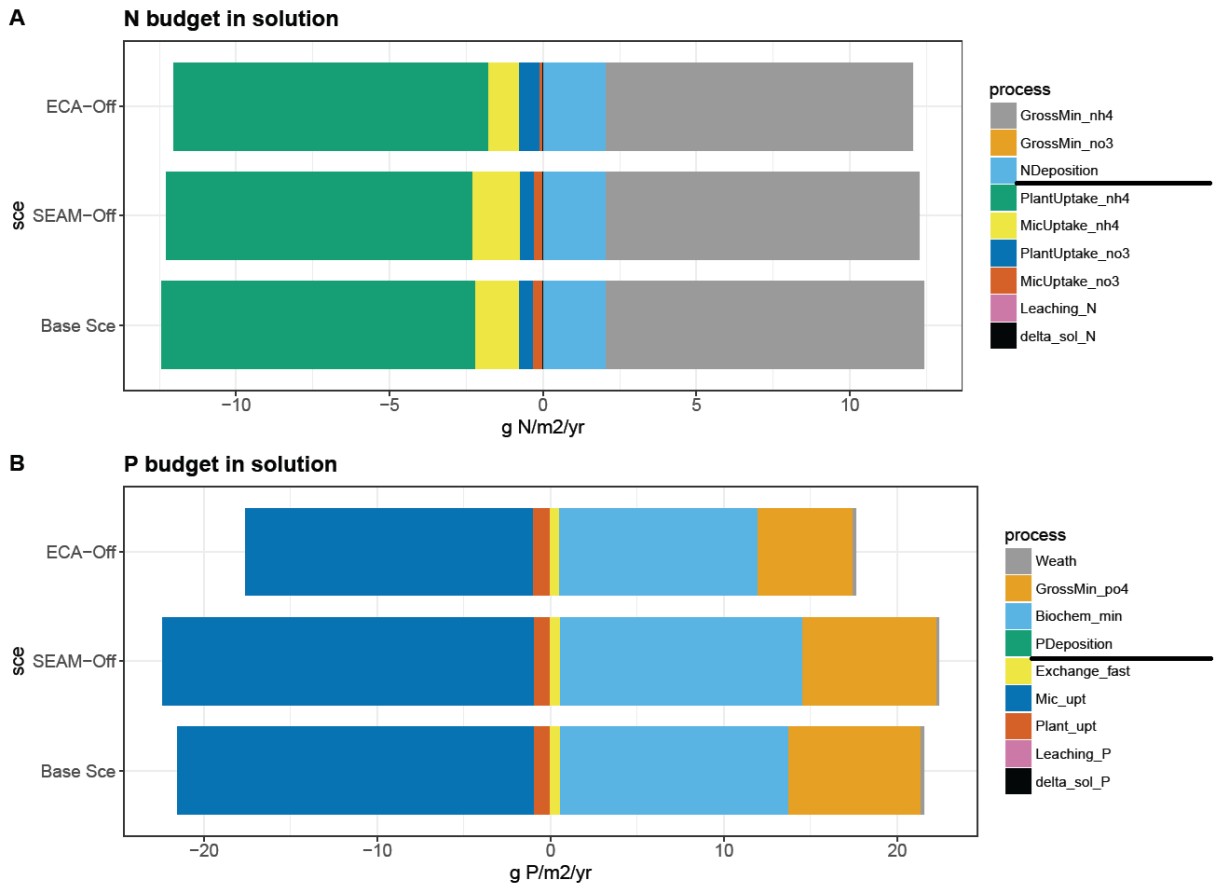

**Figure 7.** Simulated yearly budget of (A) N and (B) P in soil solutions. In panel A, sourcing fluxes of N are presented in the order of gross mineralisation of $NH_4$ and $NO_3$, N deposition (In the bar plot: from right to the zero point; in the legend: from the top to the separation line); sinking fluxes of N are presented in the order of plant and microbial uptakes of $NH_4$, plant and microbial uptakes of $NO_3$, N leaching (both inorganic and organic) and changes in soluble N content ($delta\_sol\_N$) (In the bar plot: from left to the zero point; in the legend: from the separation line to the bottom). In panel B, sourcing fluxes of P include weathering, gross mineralisation of $PO_4$, biochemical mineralisation of $PO_4$ and P deposition; sinking fluxes of P includes adsorption ($Exchange\_fast$), microbial and plant uptakes, P leaching (both inorganic and organic) and changes in soluble P content ($delta\_sol\_P$) (The order of presented processes follows the same rule as N). The budgets are calculated using data from the full simulation (200 years) of the model experiments.

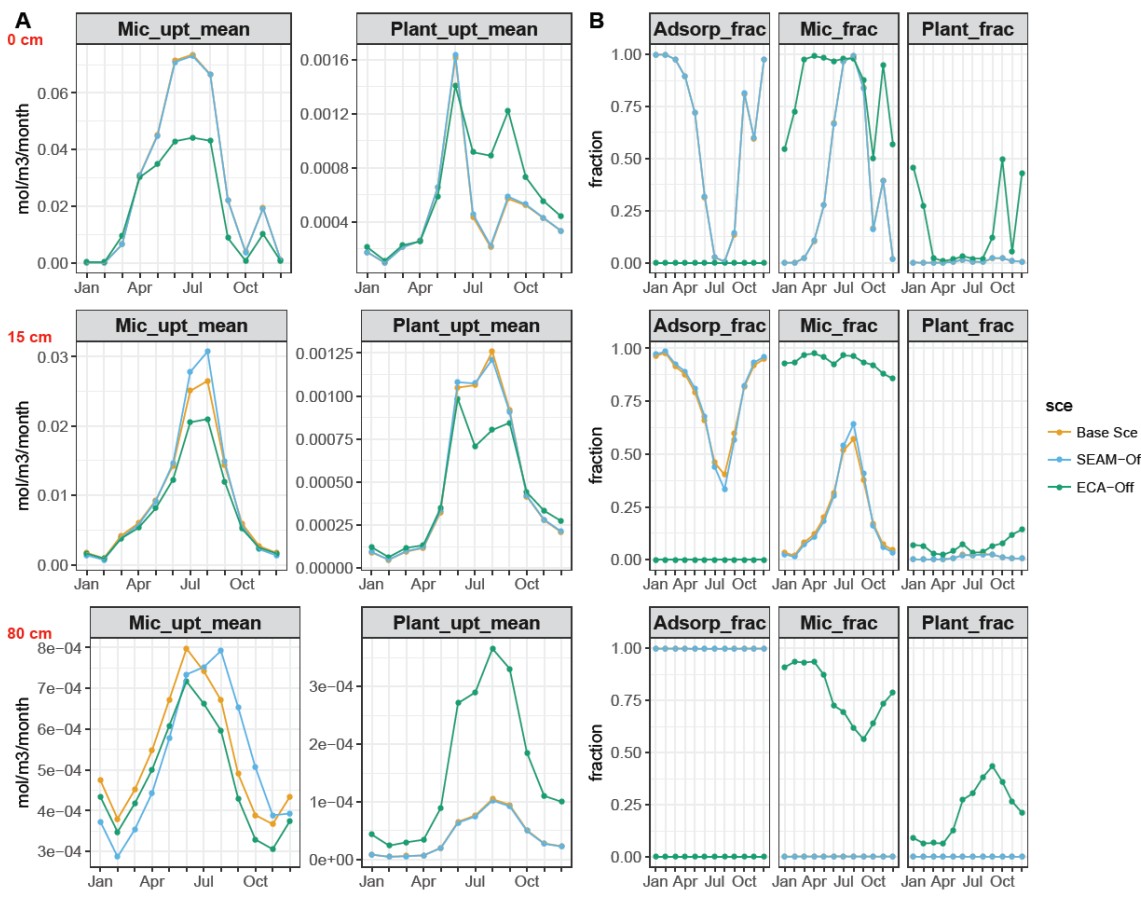

**Figure 8.** Simulated (A) microbial and plant P uptake rates and (B) relative competitiveness (in fractions) of P adsorption, microbial P uptake and plant P uptake at depths of 0 (O–A horizon, upper panels), 15 (A–B horizon, middle panels), and 80 cm (B–C horizon, bottom panels). In panel (A), monthly mean values at different depths are presented throughout whole year; in panel (B), relative competitiveness is calculated as the fraction of the individual rate to the sum of all three rates (P adsorption rate, microbial P uptake and plant P uptake). All data points are derived from data from the last 10 years of model experiments.

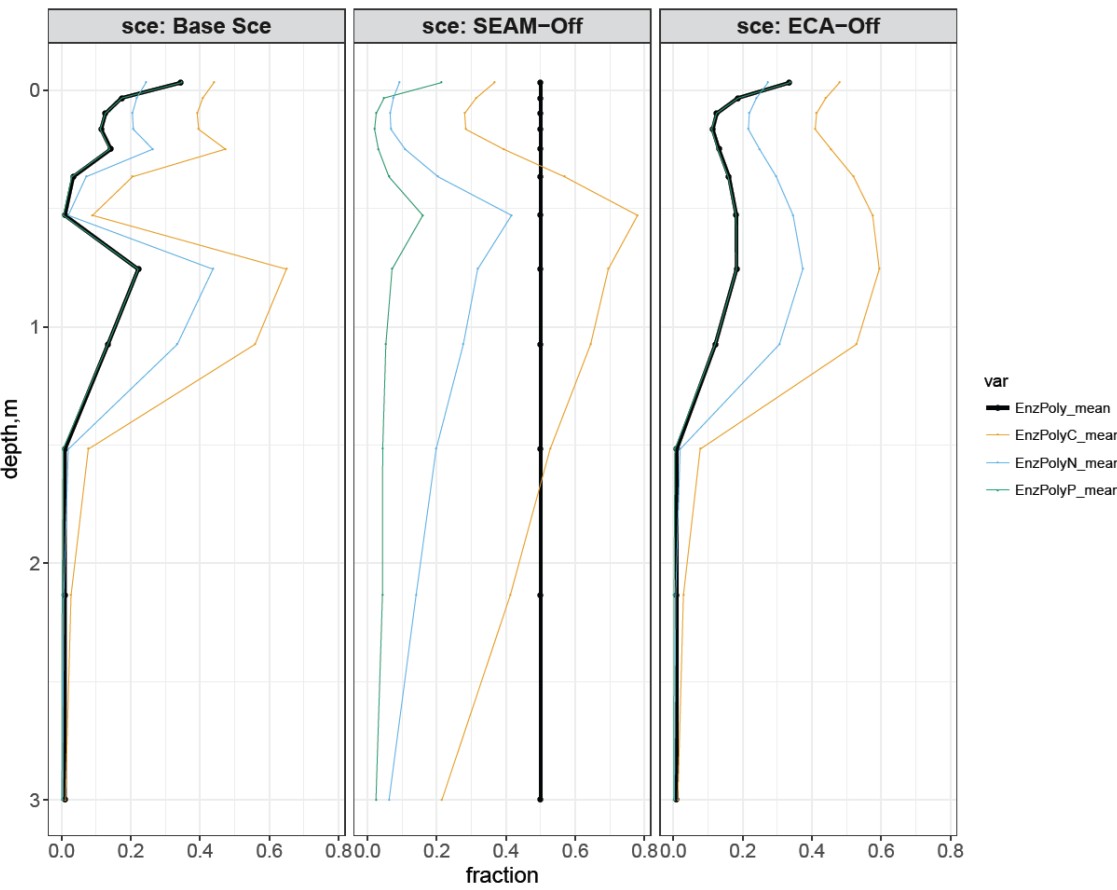

**Figure 9.** Simulated enzyme allocation to polymeric litter compared with potential enzyme allocations to polymeric litter given that the element C/N/P is the most limiting. Black lines: fraction of enzyme allocated to polymeric litter; orange/blue/green lines: potential enzyme allocation to polymeric litter to maximise C/N/P, respectively. Left panel: base scenario, middle panel: SEAM-off scenario, right panel: ECA-off scenario. All data points are derived from data from the last 10 years of model experiments.

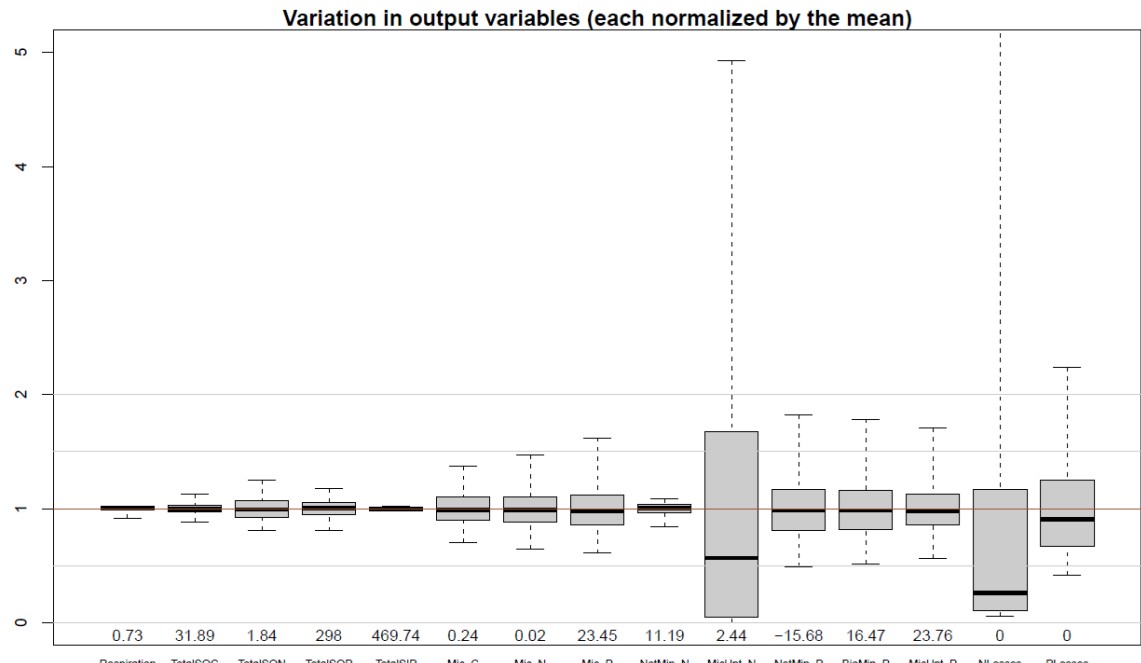

**Figure 10.** Normalised output variations in the LHS sensitivity analysis. The selected output variables include respiration; total soil organic C, N and P; microbial C, N and P; net N mineralisation; microbial N uptake; net P mineralisation; biomineralisation of P; microbial P uptake and N and P losses. All the calculations are performed for the topmost 1 m of soil based on data from the last 10 years of 1000 LHS simulations.

**Table 1.** The annual soil C, N and P fluxes of model experiments at the study site. Positive values infer accumulation in the soil, and negative values infer loss from the soil. The values are the accumulated sum of the whole soil profile, calculated based on data from the last 10 years of model experiments.

| Variable | Unit | Base Scenario | SEAM-Off | ECA-Off | 1000y | 5000y | 10000y |
|---|---|---|---|---|---|---|---|
| **Biogeochemical fluxes** | | | | | | | |
| C litterfall | $gC\,m^{-2}\,yr$ | 788.0 | 788.0 | 788.0 | 788.0 | 788.0 | 788.0 |
| Respiration | $gC\,m^{-2}\,yr$ | -741.0 | -746.2 | -746.2 | -778.0 | -787.4 | -788.0 |
| $\Delta SOC$ | $gC\,m^{-2}\,yr$ | 47.0 | 41.8 | 41.8 | 10.1 | 0.7 | 0.04 |
| N litterfall | $gN\,m^{-2}\,yr$ | 14.52 | 14.52 | 14.52 | 14.52 | 14.52 | 14.52 |
| N deposition | $gN\,m^{-2}\,yr$ | 2.04 | 2.04 | 2.04 | 2.04 | 2.04 | 2.04 |
| Plant N uptake | $gN\,m^{-2}\,yr$ | -13.29 | -13.26 | -13.28 | -15.67 | -16.01 | -16.01 |
| N leaching | $gN\,m^{-2}\,yr$ | -0.01 | -0.01 | -0.01 | -0.08 | -0.49 | -0.54 |
| $\Delta SON$ | $gN\,m^{-2}\,yr$ | 3.25 | 3.29 | 3.26 | 0.80 | 0.06 | 0.002 |
| P litterfall | $mgP\,m^{-2}\,yr$ | 980.4 | 980.4 | 980.4 | 980.4 | 980.4 | 980.4 |
| P deposition | $mgP\,m^{-2}\,yr$ | 4.2 | 4.2 | 4.2 | 4.2 | 4.2 | 4.2 |
| P weathering | $mgP\,m^{-2}\,yr$ | 155.6 | 155.6 | 142.6 | 277.8 | 197.0 | 522.8 |
| Plant P uptake | $mgP\,m^{-2}\,yr$ | -852.0 | -866.8 | -886.9 | -920.9 | -959.5 | -1134.6 |
| P leaching | $mgP\,m^{-2}\,yr$ | -0.3 | -0.3 | -1.7 | -0.5 | -1.7 | -8.4 |
| P desorption | $mgP\,m^{-2}\,yr$ | -233.0 | -243.8 | -185.6 | -58.3 | 157.4 | 345.7 |
| $\Delta SOP$ | $mgP\,m^{-2}\,yr$ | 520.9 | 516.9 | 424.1 | 399.3 | 63.0 | 18.7 |

**Table 2.** The five most important parameters (Par) and their respective RPCCs for each output variable and the overall model importance (OVI). RPCCs were calculated for each output variable, and the overall importance of parameters was measured by calculating the mean of the absolute RPCCs across all output variables, weighted by the uncertainty contribution of these model outputs. The parameters are listed in Tab.S2 and explained in Tab.S1.

| | Rank 1 | | Rank 2 | | Rank 3 | | Rank 4 | | Rank 5 | |
|---|---|---|---|---|---|---|---|---|---|---|
| Variable | Par | RPCC | Par | RPCC | Par | RPCC | Par | RPCC | Par | RPCC |
| Total SOC | $v_{max,depoly}^{poly}$ | -0.84 | $v_{max,depoly}^{res}$ | -0.80 | $\frac{1}{\tau_{mic}}$ | 0.83 | $\eta_{C,wl\to poly}$ | 0.71 | $\eta_{C,sol\to dom}$ | 0.66 |
| Total SON | $\chi_{mic}^{C:N}$ | -0.99 | $v_{max,depoly}^{res}$ | -0.94 | $\eta_{C,sol\to dom}$ | 0.84 | $\chi_{mic}^{N:P}$ | 0.40 | $\eta_{C,wl\to poly}$ | 0.38 |
| Total SOP | $\chi_{mic}^{C:N}$ | -0.97 | $\chi_{mic}^{N:P}$ | -0.97 | $v_{max,biomin}$ | -0.84 | $\eta_{res\to dom}^{P}$ | -0.78 | $\frac{1}{\tau_{mic}}$ | 0.74 |
| Total SIP | $k_{weath}$ | -0.58 | $\eta_{res\to dom}^{P}$ | 0.57 | $v_{max,biomin}$ | 0.47 | $\chi_{mic}^{N:P}$ | 0.45 | $\eta_{res\to dom}^{P}$ | -0.42 |
| Microbial C | $\frac{1}{\tau_{mic}}$ | -0.98 | $\eta_{C,sol\to dom}$ | 0.86 | $\chi_{mic}^{C:N}$ | 0.68 | $\chi_{mic}^{N:P}$ | 0.67 | $\eta_{C,wl\to poly}$ | 0.67 |
| Microbial N | $\frac{1}{\tau_{mic}}$ | -0.97 | $\chi_{mic}^{C:N}$ | -0.95 | $\eta_{C,sol\to dom}$ | 0.83 | $\chi_{mic}^{N:P}$ | 0.63 | $\eta_{C,wl\to poly}$ | 0.62 |
| Microbial P | $\frac{1}{\tau_{mic}}$ | -0.96 | $\chi_{mic}^{N:P}$ | -0.94 | $\chi_{mic}^{C:N}$ | -0.93 | $\eta_{C,sol\to dom}$ | 0.79 | $\eta_{C,wl\to poly}$ | 0.55 |
| Respiration | $\frac{1}{\tau_{mic}}$ | -0.71 | $\chi_{mic}^{N:P}$ | 0.69 | $\chi_{mic}^{C:N}$ | 0.65 | $v_{max,depoly}^{res}$ | 0.45 | $mic_{cue}^{min}$ | -0.37 |
| Net N mineralisation | $\chi_{mic}^{C:N}$ | 0.97 | $v_{max,depoly}^{res}$ | 0.65 | $mic_{cue}^{min}$ | -0.40 | $\eta_{C,sol\to dom}$ | -0.32 | $\frac{1}{\tau_{mic}}$ | -0.29 |
| Microbial N uptake | $mic_{nue}$ | -0.98 | $\chi_{mic}^{C:N}$ | -0.90 | $\eta_{C,sol\to dom}$ | 0.75 | $\eta_{C,wl\to poly}$ | 0.38 | $\chi_{mic}^{N:P}$ | 0.21 |
| Net P mineralisation | $\eta_{res\to dom}^{P}$ | 0.94 | $\chi_{mic}^{N:P}$ | 0.84 | $\chi_{mic}^{C:N}$ | 0.84 | $\eta_{C,sol\to dom}$ | -0.67 | $\eta_{C,wl\to poly}$ | -0.53 |
| P Biomineralisation | $\eta_{res\to dom}^{P}$ | -0.94 | $\chi_{mic}^{N:P}$ | -0.85 | $\chi_{mic}^{C:N}$ | -0.84 | $\eta_{C,sol\to dom}$ | 0.67 | $\eta_{C,wl\to poly}$ | 0.54 |
| Microbial P uptake | $mic_{pue}$ | -0.91 | $\chi_{mic}^{N:P}$ | -0.90 | $\chi_{mic}^{C:N}$ | -0.89 | $\eta_{res\to dom}^{P}$ | -0.85 | $\eta_{C,sol\to dom}$ | 0.70 |
| N Losses | $\chi_{mic}^{N:P}$ | 0.72 | $\frac{1}{\tau_{mic}}$ | -0.72 | $\chi_{mic}^{C:N}$ | 0.67 | $v_{max,upt}^{dom}$ | 0.41 | $v_{max,biomin}$ | 0.35 |
| P Losses | $v_{max,upt}^{dom}$ | 0.22 | $mic_{pue}$ | 0.15 | $mic_{cue}^{min}$ | -0.14 | $\eta_{res\to dom}^{P}$ | -0.11 | $k_{enz,mic}^{P}$ | -0.55 |
| OVI | $\chi_{mic}^{C:N}$ | 0.73 | $\chi_{mic}^{N:P}$ | 0.57 | $\frac{1}{\tau_{mic}}$ | 0.47 | $\eta_{C,sol\to dom}$ | 0.42 | $\eta_{res\to dom}^{P}$ | 0.35 |