# Peer review of "Jena Soil Model (JSM v1.0; revision 1934): A microbial soil organic carbon model integrated with nitrogen and phosphorus processes"

_Geoscientific Model Development, 2019_

## Referee Comment (RC1) · Anonymous Referee #1 · 27 Oct 2019

Yu et al reported the development and evaluation of the microbially-explicit SOM BGC model Jena Soil Model at a temperate beech forest stand. The model was found able to reasonably reproduce the measured profile of SOM stocks and radiocarbon. It also explained why microbial residue plays an important role in SOM cycling. Further, the nutrient dynamics resulting from plant-microbial interactions simulated by the model appeared reasonable, although important nitrification-denitrification dynamics are missing. Overall, I found the paper interesting and generally well written. I think the paper will become a good read provided the authors address the following comments.

In section 2.3, subsection model protocol and calibration. I followed the authors without

any problem on the model initialization, however, it is unclear how the 200 years are aligned with the time. Did the model pretend to start from 1850? Also the 14C of litter input in last 60 years was mentioned to match the observed 14CO2 atmospheric pulse, how was this done exactly? Further, I think the inorganic P pool from Yang et al. (2013) is closer to contemporary (say year 2000) than 1850. Was this criterion appropriate? I have no answer to this last question myself, and we also struggled when doing the P cycle in our TBM. Nonetheless, I would like to know more about the authors' opinion on this.

Another question is how the SOM 14C profile is initialized? It is not very clear from current description.

In the model formulation, I saw nitrate was part of the N dynamics. However, I did not see any description of other N related biogeochemistry. My impression is that the model does not have a nitrification-denitrification process. Is this why no abiotic ammonium adsorption is considered in the model?

Further, the model predicted a number of interesting features, such as the importance of microbial residue, and that root input will result in different depolymerization dynamics. Given one purpose of modeling is to inform new empirical experiments, I think the authors can make the paper more interesting by explicitly asking what new experiments will help constrain their model.

Finally, I think the English of the paper should be further improved. I collected some of these problems below, but I recommend the authors do a more thorough check.

Other comments:

P1 Line 3, remove the redundant "potential" from "predict potential future climate feedbacks".

P1 Line 14, remove "of" from "ample of".

P1 Line 17, replace "major nutrients" with "macronutrients".

P1 Line 24, replace "reproduce the response" with "reproduce the ecosystem response".

P2 Line 1, replace "their representation" with "their poor representation".

P2 Line 2, please be specific about what "plant uptake".

P2, Line 5, remove "the" from "In these models, the nutrient".

P2, Line 7, expand "the CENTURY approach" into "the sufficiency of the CENUTRY approach".

P2, Line 8, remove "the representation of".

P2, Line 10, "one other important limitation" is awkward, please consider revision. And replace "most of the current SOM" with "most current SOM".

P2, Line 20, the sentence reads a little bit awkward, please consider revision.

P2, Line 30, remove "this" from "this competition". Also, the sentence seems incomplete, even though it is syntactically correct.

P2, Line 33, remove "for representing them".

P3, Line 2. "kinetic" should be "kinetics".

P3, line 6, replace "cycle process" with "cycling process".

P3, line 13, remove "and was"

P3, line 17, replace "a maximum" with "the maximum".

P3, line 18, add "while" before "the mathematical".

P3, line 19, replace "of the QUINCY" with "QUINCY".

P3, Line 20, replace "can be" with "can either be".

P4, line, 24, "a loam topsoil" should be "a loamy topsoil".

P4, line 27, is the unit "g/kg" meaning "g C/kg soil"?

P5, line 3, replace "the observations" with "observations".

P5, line 19, replace "we assumed increased" with "we increased".

P6, line 5, replace "the model experiments" with "model experiments".

P6, line 18, Table S4 should be "S2".

P7, line 10-11, the sentence is hard to understand due to unclear definition of organic P and stocks. Does this mean include all P from all organic SOM pools? Nor the definition of stocks is clear. Please define them clearly.

P7, line 14, perhaps Fig. 7 and Fig. 3 should be swapped, so the paper's logical flow is more continuous.

P8, line 24, remove "the fact"

P8, line 27-34, I think "actual enzyme allocation" is not a proper name here because you don't know what is happening in reality. Perhaps a better name is needed.

P9, line 9, maybe "resistant" should be replaced with a more appropriate word.

P10, line 23, replace "The fact that" with "that".

P11, line 13, perhaps "N&P" should be replaced with "N and P" for it to be consistent with the writing style of the paper. Similar changes should be made in other places.

P11, line 13, "resulted" should be "resultant".

Fig 5, some red annotation of depth overlapped with the y-stick label.

For all figures, some annotation text should use large font size, because they may become unreadable when included in the published version.

---

## Referee Comment (RC2) · Anonymous Referee #2 · 28 Oct 2019

This manuscript describes the Jena Soil Model, a new soil organic matter model that includes microbial processes, mineral sorption of organic matter, and vertically-resolved soil processes. I thought overall the manuscript was well-written, clear, and easy to follow, and the model integrates new methods for simulating microbial and mineral influences on carbon and nutrient cycling and will be a useful contribution to the biogeochemical modeling field. The introduction did an excellent job of describing the relevant issues and the context for the model. The description of the model was generally clear, although most of the details were left in supplemental material. I do have a few suggestions of areas where the clarity of the manuscript could be improved.

[Figure]

I think some additional detail about the sources of the measurements that the model was driven with and compared to would be helpful for understanding the results. The site description only covers the characteristics of the site itself (vegetation and soil types, and some soil profiles) and does not include what kind of data collections were available and the methods used to collect key data resources such as C, N, and P profiles and meteorological data. Some presentation of seasonally-varying factors such as soil moisture, temperature, and litter inputs would help with interpretation of the simulated seasonal cycles. While some of these data collections are presumably described in detail in other publications, a summary in the methods section (an expansion of section 2.2) would help make the measurement context of the simulations clearer.

The description of model processes in the text is quite short and is very focused on a few details about stoichiometry and enzymatic processes. There is a lot of detail in the model equations (in supplemental material) that is not explained in the main text. I think some expansion of the process explanation would help readers to understand some of the results. In particular, the seasonal cycles of fluxes shown in Figures 3-5 are largely controlled by moisture and temperature functions, and possibly by the seasonal phenology of vegetation forcing in model simulations, which are not explained in the text.

Specific comments:

Page 1, Line 5-6: Some microbial-explicit decomposition models have included nutrient cycle coupling for example, Abramoff et al., 2017; Sulman et al., 2017; Huang et al, 2018.

Page 2, Line 31-32: Likewise, there are some TBMs that have included more mechanistic SOM cycling and there are some microbial SOC models that include nutrient cycling.

Page 4, line 13: The "See Sect. 5" may be a mistake. Section 5 is the Conclusions. I think this should be SI section 5? Also, I would suggest explaining these processes in

more detail in the main text rather than referring readers to the complex set of equations to understand how the model works.

Page 4, line 21: "DFG" should be spelled out or defined

Page 4, line 27: "C content of SOM" is a bit confusing as it could suggest that SOM has been separated from bulk soil and the C content of only organic matter has been determined. Based on the numbers, I think this is C content of the bulk soil in those layers. I would just say "soil C content"

Page 5, lines 26-30: It's not clear from the description whether calibration was an iterative processes. Was this two-step process repeated until results were satisfactory? Was there a particular statistical method used to assess how well the model fit the data?

Page 7, lines 18-25: Since 14C measurements were an important part of the model evaluation, with some interesting interpretations, I would suggest moving the 14C comparison figure to the main text.

Page 7, lines 30-31: Were there changes in microbial growth rates over the season that could explain changes in microbial N demand? I also would suggest adding some explanation for the large spike in microbial N uptake in November. Is this something to do will autumn litterfall, like a short-term increase in N immobilization due to deposition of a large amount of fresh litter?

Page 8, line 10: What does "TW" mean?

Page 8, line 27-page 9, line 5: I had trouble following this explanation of the figure, particularly how the potential allocation curves were calculated and how they should be interpreted.

Page 9, line 7-8: Microbial N uptake and N losses were not centered around the mean. And there is no Table S4, only S1 and S2.

[Figure]

Page 10, lines 4-9: This seems like an important part of the model structure and results, and should be introduced earlier than the Discussion section. I think this modification to the model should be described in the methods. And since making the parameter depth-dependent makes a difference to the results, it might make sense to include it as a separate set of model simulations (as with the SEAM-off and ECA-off simulations) so its effect could be shown.

Page 10, lines 23-24: At steady state, plant N and P uptake would have to be close to litterfall inputs, unless there were large losses due to leaching or other loss pathways.

Page 11, Lines 7-8: Is the fact that plants mainly take up N and not mineralized P specific to this ecosystem? In a more P-limited ecosystem, would the results differ?

Page 11, line 11-12: The global microbial stoichiometry simulations should be described in the methods.

Figure 1: It would be helpful if the notation in this figure matched the notation in the equations in supplementary material.

Figure 8: This figure is difficult to understand because there is not a clear explanation of what the different variables mean.

References:

Abramoff, R.Z., Davidson, E.A. & Finzi, A.C. (2017). A parsimonious modular approach to building a mechanistic belowground carbon and nitrogen model. J. Geophys. Res. Biogeosciences, 122, 2418–2434.

Huang, Y., Guenet, B., Ciais, P., Janssens, I.A., Soong, J.L., Wang, Y., et al. (2018). ORCHIMIC (v1.0), a microbe-mediated model for soil organic matter decomposition. Geosci. Model Dev., 11, 2111–2138.

Sulman, B.N., Brzostek, E.R., Medici, C., Shevliakova, E., Menge, D.N.L. & Phillips, R.P. (2017). Feedbacks between plant N demand and rhizosphere priming depend on

type of mycorrhizal association. Ecol. Lett., 20, 1043–1053.

---

## Referee Comment (RC3) · William Wieder (Referee) · 5 Nov 2019

General comments

Yu and coauthors present a conceptually robust model that looks at soil biogeochemical processes that explicitly represents microbial activity and CNP stoichiometry in a vertically resolved model.

The work presented here does a very thorough job documenting the model configuration and performance at a well-studied site. What's less clear is why it matters? A few suggestions are described in the specific comments below.

My other major concern with the model is that it doesn't reach steady state equilibrium, instead soil C pools are accumulating at a rate that's roughly 5% of NPP (Table 1). It seems longer spin up times were tried, but since results aren't presented I'm assuming this issue persists, if so, what do soil CNP profiles look like after 10ˆ4 years, do they still match observations well? If the model just has long-term oscillations this may be less of a concern than a constant drift (as I currently understand). The spin up issues, however, seems like a significant issue that has to be addressed if models that more explicitly represent microbial activity and coupled biogeochemical cycles are ever going to be applied in TBMs, as seems to be the aim of this work. The 'lack of plant feedbacks' argument seems unsupported. Moreover, I don't really understand why / how constant 'loss' of P into 'occluded pools affect the C dynamics simulated belowground?

This spin-up issue is also one I don't know how to handle in review and my overall assessment of this work. For this reason I'm signing this review and welcome an open conversation with the authors on this concern. I appreciate all the effort that the authors have made do make a very interesting contribution to this line of work- but a model that never really reaches steady state seems very challenging to use for more than short term-studies and sites where the model can be adequately parameterized. This may be the aim of this research group, but it seems unlikely given the introduction, conclusion, and history of strong work from this research group looking at global scale C and nutrient responses for climate change projections.

Specific comments

In my opinion there's a bit too much emphasis in the introduction in playing up the novelty of this work. This is not the first model to think about vertical resolution, microbes, nutrients or ECA. It may be the first to do all these together, which can be stated, but then move on. The current review of the literature is nice, but I'd encourage the authors to avoid language that's unnecessarily dismissive of previous work.

To address my first issue of 'why this work matters' I can three of three options to

consider:

1. Idealized experiments: While the justification for including nutrients and microbial feedbacks in a model like the Jena soil model is well established in the abstract and first paragraphs of the introduction I also fear it sets up somewhat unrealistic expectations for readers. Notably, none of the results presented illustrate how the model may respond to environmental perturbations. I'm not suggesting these have to be compared to results, but instead simple idealized experiments that illustrate how the different model configurations respond to increases in litterfall inputs, root exudates, warming, or changes in precipitation. 2. Model validation: Alternatively, it seems lots of data were needed to initialize the model. This is fine for development, but how well does the model do simulating other sites? Are there other well studied sites that can be used for independent model validation? I realize this potentially an objective for future work, but it seems like typical activity for model development papers (especially in GMD) that would help illustrate the broader generalizability of the approach outlined here? 3. Sensitivity analysis: A third alternative would be to consider illustrating model sensitivities to initial conditions? Much like the idealized experiment suggestion (above), I kept finding myself wondering how sensitive the model behaves to initial conditions that are being input to the model (e.g. litterfall and microbial stoichiometry, soil texture / mineralogy, water fluxes and temperature profiles). The parameter sensitivity analysis is nice, what about other assumptions that are being made regarding inputs to what seems like a highly parameterized model? This would open up the discussion for consideration of how to run JSM in regional or global simulations (clearly the intent), where we have less certainty of how the define these characteristics (especially with multiple elements and with depth).

The authors have actually done #3 with the microbial stoichiometry section that squeezed into the discussion. Maybe the most direct path forward to satisfy this concern would be to actually flush out these findings in the methods and results (see technical comment below).

Technical corrections

Page 2, Line 10, I might include Lehmann and Kleber 2015 here.

Page 2, Line 11, Vertically resolved models are becoming more common (McGuire et al. 2018)

Page 2, Line 18, I'm not sure the assertion (made here and in the following paragraph) that microbial explicit models don't represent coupled biogeochemical cycles is accurate (Averill & Waring 2017; Schimel & Weintraub 2003; Sistla et al. 2014; Sulman et al. 2017, 2019).

Page 3, Line 4. I'm pretty sure the ECA approach is applied in E3SM land model, which I wouldn't call a prototype model.

Page 3, line 16. & Page 4. Where's section 5?

Methods. I know COMISSION already has radiocarbon, but should there be any focus on documenting how JSM implements radiocarbon in the text or appendix?

Page 7 and Fig 2 the model calculates its own bulk density?! That's pretty interesting, should this be described in the methods?

Page 7, Line 10-15. It seems odd to jump from presentation of Fig 2 to 7. Should the display items reflect the order that information is covered in the text?

Throughout, display items should be numbered in the order they are introduced in the text.

Fig 8 and Table 1 are never referenced in the results, should they be? I'd prefer these display items not be first introduced in the discussion of the findings of this study.

Fig. 7 Bottom panels of should be % modern. I also couldn't help but notice that you just have 14C data for the site. Why not run the model for longer and show result, or put the radiocarbon observations up on the plot shown here even if they're just illustrative

for 14MOC (which should be most of what makes up the bulk 14C values at depth?

Wait, the 14C data are presented in the SI (page 7, line 20- sorry I'm on a plane and don't have access to the SI material). It seems this would be a powerful constraint for the model to try and hit (and should be included in the main text). I'm struck that we can learn a good deal about the model, even if the model is not able to match radiocarbon profiles! If longer spin-up runs have already been done I can't think of any reason not to compare results to observations where they are available.

Figs 3-4, Page 7. From the text it sounds like there are observations of soil nutrient transformation (at least N mineralization). If so, can these be included on the appropriate panels, or am I misunderstood?

Page 8, line 10, what is TW in JSM?

Section 3.2. Is the strong microbial competition for P (and not N) caused by the C:N:P ratios that are prescribed for the site (and notably skewed).

Page 8, line 24, the difference among models mentioned here regarding depth profiles of N-mineralization is not obvious, at least to my eye. Regardless, avoid using 'significant' when no statistical results are presented.

Page 8, line 27, it's not clear from the methods how the actual and potential enzyme allocation curves are being calculated from the methods, or did I miss this description. I'm also still hung up on how or why this is being done if the model doesn't explicitly represent enzymes (by the way this decision not to explicitly represent enzymes makes sense to me from a purely practical / numeric standpoint)

Page 8 line 33, if P depolymerization is completely demand driven why is microbial P uptake so much lower in the ECA-off simulations (Fig 4a)? I thought these were supposed to be the 'demand based' simulations (methods)? Please clarify.

Page 9, line 5. Reference Fig 7 here?

Page 9, Line 25 these values are for soil stoichiometry? Also, what are N:P ratios for soils? Finally, to my eye it looks like the model may overestimate observed soil C:N ratios in upper soil horizons (Fig 2). Regardless, it's likely helpful to point to this display item to support claims made about soil C pools and stoichiometry made here.

Discussion: I have to admit I haven't thought much about the dynamics driving declines in soil C:P ratios with depth, nor am I very familiar with this literature. For everything the model is doing here, this text strikes me as an odd choice to highlight at the beginning of the discussion. That said, it. Is interesting. One detail I don't really follow is that to capture observations it seems like the P recycling term in the model has to be greatly reduced in model. It doesn't seem to logically follow that the community somehow shifts to 'nutrient rich' community that's also has lower nutrient use efficiency? Instead I think the findings of Rousk and Frey suggest that substrate quality determines the microbial communities in forest soils, but doesn't speak much to vertical distribution of microbes (or their stoichiometry) being. Discussed here?

Table 1 should include soil C, N, & P pools of the model after spin up, as it's hard to assess total pool sizes from figures.

Fig 8 can colors of processes in the legend match the order they are displayed on the figure. As currently presented it's not easy for readers to interpret the figure. Introduction of the microbial stoichiometry part of the discussion seems like a nice sensitivity test of the model, but I don't like this being squeezed into the discussion and SI. Why not at least justify this experiment in the methods and describe findings in the results before discussing the findings? (It also likely makes sense to keep the figures in SI). Page 12, line 25. What observations are the model able to reproduce? Can they be illustrated on the display items (* that also should be referenced here)?

Page 12, line 28, why not cite a commission paper that's already published

References:

[Figure]

Averill, C. and Waring, B.: Nitrogen limitation of decomposition and decay: How can it occur?, Glob. Chang. Biol., (June), 1–11, doi:10.1111/gcb.13980, 2017.

Lehmann, J., & Kleber, M. (2015). The contentious nature of soil organic matter. Nature, 528(7580), 60-68. doi:10.1038/nature16069

McGuire, A. D., Lawrence, D. M., Koven, C., Clein, J. S., Burke, E., Chen, G., . . . Zhuang, Q. (2018). Dependence of the evolution of carbon dynamics in the northern permafrost region on the trajectory of climate change. Proceedings of the National Academy of Sciences. doi:10.1073/pnas.1719903115

Schimel, J. P., & Weintraub, M. N. (2003). The implications of exoenzyme activity on microbial carbon and nitrogen limitation in soil: a theoretical model. Soil Biology & Biochemistry, 35(4), 549-563. doi:10.1016/S0038-0717(03)00015-4

Sistla, S. A., Rastetter, E. B. and Schimel, J. P.: Responses of a tundra system to warming using SCAMPS: A stoichiometrically coupled, acclimating microbeplantsoil model, Ecol. Monogr., 84(1), 151–170, doi:10.1890/12-2119.1, 2014.

Sulman, B. N., Brzostek, E. R., Medici, C., Shevliakova, E., Menge, D. N. L., & Phillips, R. P. (2017). Feedbacks between plant N demand and rhizosphere priming depend on type of mycorrhizal association. Ecology Letters, 20(8), 1043-1053. doi:10.1111/ele.12802

Sulman, B. N., Shevliakova, E., Brzostek, E. R., Kivlin, S. N., Malyshev, S., Menge, D. N. L. and Zhang, X.: Diverse Mycorrhizal Associations Enhance Terrestrial C Storage in a Global Model, Global Biogeochem. Cycles, 33(4), 501–523, doi:10.1029/2018GB005973, 2019.

---

## Short Comment (SC1) · 7 Nov 2019

Dear authors,

in my role as Executive editor of GMD, I would like to bring to your attention our Editorial version 1.2:

https://www.geosci-model-dev.net/12/2215/2019/

This highlights some requirements of papers published in GMD, which is also available on the GMD website in the 'Manuscript Types' section: http://www.geoscientific-model-development.net/submission/manuscript_types.html

In particular, please note that for your paper, the following requirement has not been met in the Discussions paper:

- "The main paper must give the model name and version number (or other unique identifier) in the title."

- Code must be published on a persistent public archive with a unique identifier for the exact model version described in the paper or uploaded to the supplement, unless this is impossible for reasons beyond the control of authors. All papers must include a section, at the end of the paper, entitled "Code availability". Here, either instructions for obtaining the code, or the reasons why the code is not available should be clearly stated. It is preferred for the code to be uploaded as a supplement or to be made available at a data repository with an associated DOI (digital object identifier) for the exact model version described in the paper. Alternatively, for established models, there may be an existing means of accessing the code through a particular system. In this case, there must exist a means of permanently accessing the precise model version described in the paper. In some cases, authors may prefer to put models on their own website, or to act as a point of contact for obtaining the code. Given the impermanence of websites and email addresses, this is not encouraged, and authors should consider improving the availability with a more permanent arrangement. Making code available through personal websites or via email contact to the authors is not sufficient. After the paper is accepted the model archive should be updated to include a link to the GMD paper.

Please provide the version number of the Jena Soil model in the title of your revised manuscript.

Additionally, please note, that GMD is encouraging authors to provide a persistent access to the exact version of the source code used for the model version presented in the paper. As explained in https://www.geoscientific-model-development.net/about/manuscript_types.html the preferred reference to this release is through the use of a DOI which then can be cited in the paper. For projects in GitHub a DOI for a released code version can easily be created using Zenodo, see https://guides.github.com/activities/citable-code/ for details.

Finally note, that according to our new Editorial (v1.2) all data and analysis / plotting scripts should be made available.

Yours, Astrid Kerkweg

––––––––––––––––––––––––––––––––

---

## Author Comment (AC1) · 20 Nov 2019

A: authors' response

Yu et al reported the development and evaluation of the microbially-explicit SOM BGC model Jena Soil Model at a temperate beech forest stand. The model was found able to reasonably reproduce the measured profile of SOM stocks and radiocarbon. It also explained why microbial residue plays an important role in SOM cycling. Further, the nutrient dynamics resulting from plant-microbial interactions simulated by the model appeared reasonable, although important nitrification-denitrification dynamics are missing. Overall, I found the paper interesting and generally well written. I think the paper

will become a good read provided the authors address the following comments.

A: we thank the reviewer for the positive comment and recognition of our work

In section 2.3, subsection model protocol and calibration. I followed the authors without any problem on the model initialization, however, it is unclear how the 200 years are aligned with the time. Did the model pretend to start from 1850? Also the 14C of litter input in last 60 years was mentioned to match the observed 14CO2 atmospheric pulse, how was this done exactly? Further, I think the inorganic P pool from Yang et al. (2013) is closer to contemporary (say year 2000) than 1850. Was this criterion appropriate? I have no answer to this last question myself, and we also struggled when doing the P cycle in our TBM. Nonetheless, I would like to know more about the authors' opinion on this.

A: we ran the model for 200 years and compare the simulated results with the present-day measurement; therefore the initialization should represent the condition of ca.1820, and the bomb pulse we mimicked occurred around 1960, which is ca. 60 years before the end of simulation. The pulse was fitted to the observed atmospheric peak, by simply modifying the 14C content of litter fall.

The inorganic P pool we used to initialize the model was the data set that Yang et al. published in 2014 (Yang X, Post WM, Thornton PE & Jain AK 2014: Global Gridded Soil Phosphorus Distribution Maps at 0.5-degree Resolution. ORNL Distributed Active Archive Center.), but we made a mistake in the reference and will revise it in the re-submission. The data set we used has no explicit temporal component, but data were nominally for the pre-industrial period ca. 1850 as recommended by the authors. So we don't think it is a problem to use it to represent the condition of 1820. As we stated in the discussion that the uncertainties in inorganic P cycling and initialization are very high. We have made some progress in reducing these uncertainties, and will hopefully publish the results in a separate study soon.

Another question is how the SOM 14C profile is initialized? It is not very clear from

current description.

A: We initialize the whole SOC profile with a 14C percent Modern value of 100% for all the carbon pools and then let the 14C values develop from there following COMISSION model (Ahrens et al. 2015). We will include the 14C initialization in the resubmission.

In the model formulation, I saw nitrate was part of the N dynamics. However, I did not see any description of other N related biogeochemistry. My impression is that the model does not have a nitrification-denitrification process. Is this why no abiotic ammonium adsorption is considered in the model?

A: Yes, the N dynamics in the current version is much simplified but will be implemented into the model in a later stage. In this paper, we mainly focus on the different roles of inorganic and organic nutrients in regulating the microbial/SOM dynamics and processes; therefore we think the simplified N processes won't alter the main conclusions of this study. However, we do realize this is an important point to mention and will clarify it in the summary section.

Further, the model predicted a number of interesting features, such as the importance of microbial residue, and that root input will result in different depolymerization dynamics. Given one purpose of modeling is to inform new empirical experiments, I think the authors can make the paper more interesting by explicitly asking what new experiments will help constrain their model .

A: Thanks for recognition of our work. We will include some implications for experiments in the resubmission. A few examples are: how the microbial carbon use efficiency will change when the nutrient availability changes? how the microbial enzyme production will respond to changes of litter input?

Finally, I think the English of the paper should be further improved. I collected some of these problems below, but I recommend the authors do a more thorough check.

A: Thanks for helping with the language. We will do a grammar check before resub-

[Figure]

mission.

Other comments:

P1 Line 3, remove the redundant "potential" from "predict potential future climate feed-backs".

A: Corrected

P1 Line 14, remove "of" from "ample of".

A: Corrected

P1 Line 17, replace "major nutrients" with "macronutrients".

A: Corrected

P1 Line 24, replace "reproduce the response" with "reproduce the ecosystem re-sponse".

A: Corrected

P2 Line 1, replace "their representation" with "their poor representation".

A: Corrected

P2 Line 2, please be specific about what "plant uptake".

A: Will be revised

P2, Line 5, remove "the" from "In these models, the nutrient".

A: Corrected

P2, Line 7, expand "the CENTURY approach" into "the sufficiency of the CENUTRY approach".

A: Corrected

P2, Line 8, remove "the representation of".

A: Corrected

P2, Line 10, "one other important limitation" is awkward, please consider revision. And replace "most of the current SOM" with "most current SOM".

A: Corrected

P2, Line 20, the sentence reads a little bit awkward, please consider revision.

A: Will be revised

P2, Line 30, remove "this" from "this competition". Also, the sentence seems incomplete, even though it is syntactically correct.

A: Will be revised

P2, Line 33, remove "for representing them".

A: Corrected

P3, Line 2. "kinetic" should be "kinetics".

A: Corrected

P3, line 6, replace "cycle process" with "cycling process".

A: Corrected

P3, line 13, remove "and was"

A: Corrected

P3, line 17, replace "a maximum" with "the maximum".

A: Corrected

P3, line 18, add "while" before "the mathematical".

A: Corrected

P3, line 19, replace "of the QUINCY" with "QUINCY".

A: Corrected

P3, Line 20, replace "can be" with "can either be".

A: Corrected

P4, line, 24, "a loam topsoil" should be "a loamy topsoil".

A: Corrected

P4, line 27, is the unit "g/kg" meaning "g C/kg soil"?

A: Yes. Corrected

P5, line 3, replace "the observations" with "observations".

A: Corrected

P5, line 19, replace "we assumed increased" with "we increased".

A: Corrected

P6, line 5, replace "the model experiments" with "model experiments".

A: Corrected

P6, line 18, Table S4 should be "S2".

A: Corrected

P7, line 10-11, the sentence is hard to understand due to unclear definition of organic P and stocks. Does this mean include all P from all organic SOM pools? Nor the definition of stocks is clear. Please define them clearly.

A: Will be clarified in the resubmission.

P7, line 14, perhaps Fig. 7 and Fig. 3 should be swapped, so the paper's logical flow is more continuous.

P8, line 24, remove "the fact"

A: Corrected

P8, line 27-34, I think "actual enzyme allocation" is not a proper name here because you don't know what is happening in reality. Perhaps a better name is needed.

A: Will be replaced with a less confusing name

P9, line 9, maybe "resistant" should be replaced with a more appropriate word.

A: Will be revised

P10, line 23, replace "The fact that" with "that".

A: Corrected

P11, line 13, perhaps "N&P" should be replaced with "N and P" for it to be consistent with the writing style of the paper. Similar changes should be made in other places.

A: Corrected. We will check the consistency of other terms in the resubmission

P11, line 13, "resulted" should be "resultant".

A: Corrected

Fig 5, some red annotation of depth overlapped with the y-stick label.

A: Will be revised

For all figures, some annotation text should use large font size, because they may become unreadable when included in the published version.

A: Will be revised

---

## Author Comment (AC4) · 20 Nov 2019

A: authors' response

Please provide the version number of the Jena Soil model in the title of your revised manuscript.

A: We will do that in the resubmission.

Additionally, please note, that GMD is encouraging authors to provide a persistent access to the exact version of the source code used for the model version presented in the paper. As explained in https://www.geoscientific-

modeldevelopment.net/about/manuscript_types.html the preferred reference to this release is through the use of a DOI which then can be cited in the paper. For projects in GitHub a DOI for a released code version can easily be created using Zenodo, see https://guides.github.com/activities/citable-code/ for details.

A: we are working to find the most suitable way for us to release the codes with a DOI.

Finally note, that according to our new Editorial (v1.2) all data and analysis / plotting scripts should be made available.

A: All the measurements used in the paper are published data. The model data and plotting scripts will be made available.

––––––––––––––––––––––––––––––

---

## Author Response (AR1)

R: reviewer's comment

*A: authors' response*

C: changes made in the manuscript (the tracked-change version)

**Responses to Reviewer Comment 1**

**R: Yu et al reported the development and evaluation of the microbially-explicit SOM BGC model Jena Soil Model at a temperate beech forest stand. The model was found able to reasonably reproduce the measured profile of SOM stocks and radiocarbon. It also explained why microbial residue plays an important role in SOM cycling. Further, the nutrient dynamics resulting from plant-microbial interactions simulated by the model appeared reasonable, although important nitrification-denitrification dynamics are missing. Overall, I found the paper interesting and generally well written. I think the paper will become a good read provided the authors address the following comments.**

*A: we thank the reviewer for the positive comment and recognition of our work*

**R: In section 2.3, subsection model protocol and calibration. I followed the authors without any problem on the model initialization, however, it is unclear how the 200 years are aligned with the time. Did the model pretend to start from 1850? Also the 14C of litter input in last 60 years was mentioned to match the observed 14CO2 atmospheric pulse, how was this done exactly? Further, I think the inorganic P pool from Yang et al. (2013) is closer to contemporary (say year 2000) than 1850. Was this criterion appropriate? I have no answer to this last question myself, and we also struggled when doing the P cycle in our TBM. Nonetheless, I would like to know more about the authors' opinion on this.**

*A: we ran the model for 200 years and compared the simulated results with the present-day measurement; therefore the initialization should represent the condition of ca.1820, and the bomb pulse we mimicked occurred around 1960, which is ca. 60 years before the end of simulation. The pulse was fitted to the observed atmospheric peak, by simply modifying the 14C content of litter fall.*

C: Page7, Line16-18: "To mimic the history of $^{14}C$ input, we increased litter $^{14}C$ content for the final 60 years before the end of the simulation, assuming that the $\Delta^{14}C$ in gross primary productivity in response to the observed $CO_2$ atmospheric pulse propagates directly into litterfall without any delay."

*A: The inorganic P pool we used to initialize the model was the data set that Yang et al. published in 2014 (Yang X, Post WM, Thornton PE & Jain AK 2014: Global Gridded Soil Phosphorus Distribution Maps at 0.5-degree Resolution. ORNL Distributed Active Archive Center.), but we made a mistake in the reference and will revise it in the resubmission. The data set we used has no explicit*

*temporal component, but data were nominally for the pre-industrial period ca. 1850 as recommended by the authors. So we don't think it is a problem to use it to represent the condition of 1820. As we stated in the discussion that the uncertainties in inorganic P cycling and initialization are very high. We have made some progress in reducing these uncertainties, and will hopefully publish the results in a separate study soon.*

C: Page7, Line9-10: "The soil inorganic P pools were initialised using the soil P dataset from Yang et al. (2014a), …"

**R: Another question is how the SOM 14C profile is initialized? It is not very clear from current description.**

*A: We initialize the whole SOC profile with a pre-industrial 14C value for all the carbon pools and then let the 14C values develop from there following COMISSION model (Ahrens et al. 2015). We will include the 14C initialization in the resubmission.*

C: Page7, Line8-9: "All SOC profiles were initialised with a pre-industrial $\Delta^{14}C$ values for all C pools, from which the $^{14}C$ values were developed."

**R: In the model formulation, I saw nitrate was part of the N dynamics. However, I did not see any description of other N related biogeochemistry. My impression is that the model does not have a nitrification-denitrification process. Is this why no abiotic ammonium adsorption is considered in the model?**

*A: Yes, the N dynamics in the current version is much simplified but will be implemented into the model in a later stage. In this paper, we mainly focus on the different roles of inorganic and organic nutrients in regulating the microbial/SOM dynamics and processes; therefore we think the simplified N processes won't alter the main conclusions of this study. However, we do realize this is an important point to mention and will clarify it in the summary section.*

C: Page18, Line15-20: "Concerning the model's description of N dynamics, in the current version, N processes such as nitrification/denitrification and abiotic ammonium adsorption are not yet implemented. Although the simplified N dynamics will probably not alter the main findings of this study, it is important to investigate these in the future since plants often have a preference for ammonium uptake (Masclaux-Daubresse et al., 2010). Finally, given the good quality of the input data, JSM could adequately reproduce the soil stocks and flux rates at the selected study site; however, its capacity to extrapolate to other climate and soil conditions needs to be further investigated in the future."

**R: Further, the model predicted a number of interesting features, such as the importance of microbial residue, and that root input will result in different depolymerization dynamics. Given one purpose of modeling is to inform new empirical experiments, I think the authors can make the paper more interesting by explicitly asking what new experiments will help constrain their model.**

*A: Thanks for recognition of our work. We will include some implications for experiments in the resubmission. A few examples are: how the microbial carbon use efficiency will change when the nutrient availability changes? how the microbial enzyme production will respond to changes of litter input?*

C: Page19, Line11-15: "To better represent microbial dynamics, we would need detailed and advanced understanding of microbial processes from experiments for implementation and testing in the model. For example, how will microbial C use efficiency change in response to changes in C sources (e.g. DOM or litter addition) and nutrient availability (e.g. N & P addition)? How starkly does the microbial community adjust its stoichiometry, change its element use efficiency or alter extracellular enzyme synthesis under dynamic external conditions?"

**R: Finally, I think the English of the paper should be further improved. I collected some of these problems below, but I recommend the authors do a more thorough check.**

*A: Thanks for helping with the language. We will do a grammar check before resubmission.*

C: The language was edited by a professional editor before resubmission. Please find it in the tracked-change version.

**Other comments:**

**R: P1 Line 3, remove the redundant "potential" from "predict potential future climate feedbacks".**

*A: this part was deleted in the resubmission.*

C: Page1, Line1-3.

**R: P1 Line 14, remove "of" from "ample of".**

*A: Corrected*

C: Page1, Line20: "There is ample evidence from both ecosystem monitoring data …"

**R: P1 Line 17, replace "major nutrients" with "macronutrients".**

*A: Corrected*

C: Page1 Line24-Page2 Line1: "… on terrestrial ecosystems are driven by the constraints imposed by macronutrients such as nitrogen (N) and phosphorus (P)"

**R: P1 Line 24, replace "reproduce the response" with "reproduce the ecosystem response".**

*A: Corrected*

C: Page2 Line8-9: "…, these nutrient-enabled TBMs largely fail to reproduce the responses of ecosystems to elevated atmospheric $CO_2$ concentration, …"

**R: P2 Line 1, replace "their representation" with "their poor representation".**

*A: Corrected*

C: Page2 Line11-12: "An important shortcoming of the current generation of models is their poor representation of plant–soil interactions, …"

**R: P2 Line 2, please be specific about what "plant uptake".**

*A: Revised*

C: Page2 Line13-14: "… to altered plant inputs and ultimately plant uptake of mineral nutrients (Hinsinger et al., 2011; Drake et al., 2011; Zaehle et al., 2014)."

**R: P2, Line 5, remove "the" from "In these models, the nutrient".**

*A: Corrected*

C: Page2 Line16-17: "In these models, nutrient mineralisation and immobilisation fluxes …."

**R: P2, Line 7, expand "the CENTURY approach" into "the sufficiency of the CENUTRY approach".**

*A: Revised to "the adequacy of"*

C: Page2 Line18-19: "Recent insights in soil science have questioned the adequacy of the CENTURY approach …"

**R: P2, Line 8, remove "the representation of".**

*A: Corrected*

C: Page2 Line21: "such as the substrate limitation of soil microbial growth …"

**R: P2, Line 10, "one other important limitation" is awkward, please consider revision. And replace "most of the current SOM" with "most current SOM".**

*A: Revised*

C: Page2 Line24-25: "Another limitation of many current SOM models in TBMs is that they represent soil as a 'bucket', …"

**R: P2, Line 20, the sentence reads a little bit awkward, please consider revision.**

*A: Revised*

C: Page3 Line3-6: "The main challenge in coupling C and nutrient cycles in microbially explicit models is to account for the large stoichiometric imbalances between the microbial decomposers (i.e. soil microorganisms) and their resources (i.e. plant litter and SOM) (Xu et al., 2013; Mooshammer et al., 2014)."

**R: P2, Line 30, remove "this" from "this competition". Also, the sentence seems incomplete, even though it is syntactically correct.**

*A: Revised*

C: Page3 Line29-30: "Regarding P, in particular, the soil mineral surface adsorbs inorganic P to compete with plants and microbes (Bünemann et al., 2016; Spohn et al., 2018)."

**R: P2, Line 33, remove "for representing them".**

*A: The sentence is rewritten*

C: Page3 Line19-24: "As the above-mentioned processes/phenomena are receiving more attentions, an increasing number of emerging microbially explicit models have started to tackle these challenges by accounting for the N cycle, enzymatic biosynthesis and rhizosphere priming (Abramoff et al., 2017; Sulman et al., 2017; Huang et al., 2018; Sulman et al., 2019) using certain novel approaches"

**R: P3, Line 2. "kinetic" should be "kinetics".**

*A: The sentence is rewritten*

C: Page3 Line30-33: "The equilibrium chemistry approximation (ECA) approach has been proposed to simulate the competition of substrate uptake kinetics in complex networks where the uptake kinetics of one substrate affects the others (Tang and Riley, 2013)."

**R: P3, line 6, replace "cycle process" with "cycling process".**

*A: The sentence is rewritten*

C: Page4 Line1-2: "…, we present the structure and basic features of a novel microbially explicit and vertically resolved SOM model that integrates with the N and P cycles—the Jena Soil Model (JSM)."

**R: P3, line 13, remove "and was"**

*A: Corrected*

C: Page4 Line13-14: "JSM is a soil biogeochemical model built on the backbone of the vertically explicit C-only SOC model COMISSION (Ahrens et al., 2015)"

**R: P3, line 17, replace "a maximum" with "the maximum".**

*A: Corrected*

C: Page4 Line14-15: "The COMISSION model was further developed from the conventional one by introducing a scalable maximum sorption capacity"

**R: P3, line 18, add "while" before "the mathematical".**

*A: Corrected*

C: Page4 Line19-20: "A schematic overview of JSM is presented in Fig. 1, and the mathematical description of the processes is provided in Appendix A."

**R: P3, line 19, replace "of the QUINCY" with "QUINCY".**

*A: Corrected*

C: Page4 Line20-21: "The model is integrated into the QUINCY (Thum et al., 2019) TBM modelling framework …"

**R: P3, Line 20, replace "can be" with "can either be".**

*A: Corrected*

C: Page4 Line22: "… and can either be applied as a stand-alone soil model or …"

**R: P4, line, 24, "a loam topsoil" should be "a loamy topsoil".**

*A: Corrected*

C: Page6 Line11: "…, with loamy topsoil and sandy loamy subsoil, …"

**R: P4, line 27, is the unit "g/kg" meaning "g C/kg soil"?**

*A: Yes. Corrected in all appearances*

C: Page6 Line20 and so on: "The soil C content decreases from 510 g C/kg soil in the forest floor to 126 g C/kg soil …"

**R: P5, line 3, replace "the observations" with "observations".**

A: Corrected

C: Page6 Line31: "… were obtained from observations at the VES site"

**R: P5, line 19, replace "we assumed increased" with "we increased".**

*A: Corrected*

C: Page7 Line17: "…, we increased litter $^{14}$C content for the final 60 years …"

**R: P6, line 5, replace "the model experiments" with "model experiments".**

*A: Corrected*

C: Page8 Line5: "All model experiments used the same parameterization …"

**R: P6, line 18, Table S4 should be "S2".**

*A: Corrected, should be "S1"*

C: Page9 Line4: "We selected 28 parameters from calibration (Tab.S1) and …"

**R: P7, line 10-11, the sentence is hard to understand due to unclear definition of organic P and stocks. Does this mean include all P from all organic SOM pools? Nor the definition of stocks is clear. Please define them clearly.**

*A: Revised and clarified.*

C: Page10 Line1-3: "The modelled results agreed well with observed stock sizes and vertical patterns, indicating that the stocks [here we define the term `stock' as the total amount of all (model) pools within a larger set] of C, N and P pools …"

**R: P7, line 14, perhaps Fig. 7 and Fig. 3 should be swapped, so the paper's logical flow is more continuous.**

*A: Corrected*

C: all the displayed items in the manuscript are re-numbered in their order of appearances.

**R: P8, line 24, remove "the fact"**

*A: The sentence is rewritten*

C: Page12 Line4: "This difference is because that geophysical processes, …"

**R: P8, line 27-34, I think "actual enzyme allocation" is not a proper name here because you don't know what is happening in reality. Perhaps a better name is needed.**

*A: Revised. The word "actual" is removed*

C: Page12 Line9-10: "We compared the enzyme allocation curve of polymeric litter …"

**R: P9, line 9, maybe "resistant" should be replaced with a more appropriate word.**

*A: Revised to "insensitive"*

C: Page12 Line29: "N mineralisation was surprisingly insensitive while …"

**R: P10, line 23, replace "The fact that" with "that".**

*A: Corrected*

C: Page14 Line24: "The simulated plant N and P uptakes …"

**R: P11, line 13, perhaps "N&P" should be replaced with "N and P" for it to be consistent with the writing style of the paper. Similar changes should be made in other places.**

*A: Corrected. We followed the advice from the language editor and use "N & P" throughout the manuscript.*

C: Page15 Line20: "…, although the N & P stocks and fluxes were greatly influenced."

**R: P11, line 13, "resulted" should be "resultant".**

*A: Corrected*

C: Page15 Line21: "…, the resultant SOM C:N and C:P ratios became lower and higher, …"

**R: Fig 5, some red annotation of depth overlapped with the y-stick label.**

*A: Revised*

C: The new figure number is Fig 8.

**R: For all figures, some annotation text should use large font size, because they may become unreadable when included in the published version.**

*A: Revised*

C: the annotation text in Fig.8 and Fig.9 are enlarged.

**Responses to Reviewer Comment 2**

**R: This manuscript describes the Jena Soil Model, a new soil organic matter model that includes microbial processes, mineral sorption of organic matter, and vertically-resolved soil processes. I thought overall the manuscript was well-written, clear, and easy to follow, and the model integrates new methods for simulating microbial and mineral influences on carbon and nutrient cycling and will be a useful contribution to the biogeochemical modeling field. The introduction did an excellent job of describing the relevant issues and the context for the model. The description of the model was generally clear, although most of the details were left in supplemental material. I do have a few suggestions of areas where the clarity of the manuscript could be improved.**

_A: we thank the reviewer for the positive comment and recognition of our work_

**R: I think some additional detail about the sources of the measurements that the model was driven with and compared to would be helpful for understanding the results. The site description only covers the characteristics of the site itself (vegetation and soil types, and some soil profiles) and does not include what kind of data collections were available and the methods used to collect key data resources such as C, N, and P profiles and meteorological data. Some presentation of seasonally-varying factors such as soil moisture, temperature, and litter inputs would help with interpretation of the simulated seasonal cycles. While some of these data collections are presumably described in detail in other publications, a summary in the methods section (an expansion of section 2.2) would help make the measurement context of the simulations clearer.**

_A: we have included a summary of the measurements in the method section to give a bit more information on the data collections._

C: Page6 Line15-18: "The soil was sampled up to 1 m, with layer depths of 5–10 cm, for the measurements of total C, N and organic and inorganic P and basic physical properties such as bulk density and soil texture. Soil from the A horizon alone was extracted for the estimation of microbial C, N and P pools. Detailed sampling and measurement approaches are described in Lang et al. (2017)."

**R: The description of model processes in the text is quite short and is very focused on a few details about stoichiometry and enzymatic processes. There is a lot of detail in the model equations (in supplemental material) that is not explained in the main text. I think some expansion of the process explanation would help readers to understand some of the results. In particular, the seasonal cycles of fluxes shown in Figures 3-5 are largely controlled by moisture and temperature functions, and possibly by the seasonal phenology of vegetation forcing in model simulations, which are not explained in the text.**

*A: we agree that the seasonal patterns are strongly controlled by the temperature and the seasonal variation of the litter forcing. Although the main focus of this paper is not to look at the causes of seasonal pattern, we do agree it is better to mention them in the method and discussion sections.*

*We have added some brief descriptions of other processes, such as the temperature and moisture sensitivities used and the microbial response to nutrient availabilities in the model description to help readers better understand our results.*

C: Page5 Line6-9: "It assimilates organic forms of C, N and P from DOM with fixed element use efficiencies and inorganic forms of N and P from soluble mineral pools. Microbes are assumed to aim to maximise their growth by maintaining high C use efficiency; however, when growth is limited by nutrients, microbes reduce their C use efficiency and increase nutrient mineralisation accordingly (See Sect.S1.5)."

C: Page5 Line31-Page6 Line2: "The impacts of soil conditions on biogeochemical processes are also represented in JSM. The temperature response of different processes (e.g. microbial growth, decay, and nutrient uptake in Sect.S1.4) are represented by Arrhenius equation with different activation energies. Moisture responses are described by two rate modifiers—one representing the effects of oxygen limitation (e.g. litter turnover in Sect.S1.2) and the other representing the effects of diffusion limitation (e.g. depolymerisation in Sect.S1.3). JSM also considers the effects of SOM content to correct bulk density (Sect.S3), which in turn affects other processes such as organic matter (Eq.S7) and phosphate (Eq.S25) sorption."

**Specific comments:**

**R: Page 1, Line 5-6: Some microbial-explicit decomposition models have included nutrient cycle coupling for example, Abramoff et al., 2017; Sulman et al., 2017; Huang et al, 2018.**

*A: Thanks for the information. We have corrected it.*

C: Page1 Line7-8: "…, they lack a full coupling to the nitrogen (N) and phosphorus (P) cycles with the soil profile."

**R: Page 2, Line 31-32: Likewise, there are some TBMs that have included more mechanistic SOM cycling and there are some microbial SOC models that include nutrient cycling.**

*A: We have corrected it.*

C: Page3 Line19-24: "As the above-mentioned processes/phenomena are receiving more attentions, an increasing number of emerging microbially explicit models have started to tackle these challenges by accounting for the N cycle, enzymatic biosynthesis and rhizosphere priming (Abramoff et al., 2017; Sulman et al., 2017; Huang et al., 2018; Sulman et al., 2019) using certain novel approaches"

**R: Page 4, line 13: The "See Sect. 5" may be a mistake. Section 5 is the Conclusions. I think this should be SI section 5? Also, I would suggest explaining these processes in**

**more detail in the main text rather than referring readers to the complex set of equations to understand how the model works.**

*A: Corrected. We will add a brief description of these processes to the model description, as mentioned in the previous response.*

C: Page5 Line24-27: "JSM tracks three potential fractions of enzyme allocation, which represent cases in which microbes only maximise depolymerisation release of C, N or P, respectively, and then updates the microbial enzyme allocation fraction by acclimating gradually to the potential fraction of most limiting element (See Sect.S1.5.2)."

**R: Page 4, line 21: "DFG" should be spelled out or defined**

*A: Corrected.*

C: Page6 Line9-10: "…, the VES site has also been one of the main study sites in the German Research Foundation (DFG) funded the priority programme 1685 …"

**R: Page 4, line 27: "C content of SOM" is a bit confusing as it could suggest that SOM has been separated from bulk soil and the C content of only organic matter has been determined. Based on the numbers, I think this is C content of the bulk soil in those layers. I would just say "soil C content"**

*A: Thanks for pointing it out. Corrected.*

C: Page6 Line20: "The soil C content decreases from 510 g C/kg soil in the forest floor to 126 g C/kg soil …"

**R: Page 5, lines 26-30: It's not clear from the description whether calibration was an iterative processes. Was this two-step process repeated until results were satisfactory? Was there a particular statistical method used to assess how well the model fit the data?**

*A: Thanks for pointing it out. No, the calibration is not done iteratively, but indeed, during the second step, we slightly revised some of the parameter values of the first step based on our previously experience.*

*Also, we only evaluated the model fit visually and did not use a particular statistical method. Because we did not run a Monte-Carlo type calibration, instead all the parameters were varied gradually between two selected values. By calibrating in this way, we learnt how the individual parameter/process would affect other processes/pools, and it also makes the visual judgment sufficient to choose the better model fit.*

C: Page8 Line21-25: "The two steps were not performed iteratively; however, during the second step, we revised the parameters from the first step as necessary. Other observed soil profiles, such as the soil organic N and the bulk density, were used as additional criteria to select parameterisation, although not specifically used to calibrate the model. During the

calibration processes, parameter values were gradually changed and the goodness of model fit was visually evaluated on the basis of observations."

**Page 7, lines 18-25: Since 14C measurements were an important part of the model evaluation, with some interesting interpretations, I would suggest moving the 14C comparison figure to the main text.**

*A: The 14C signal is indeed a very important feature of our model, but we did not include the comparison in the main text for two reasons: first, the main focus of this paper is to include nutrient cycles and discuss the features more relevant with carbon-nutrient interactions; second, we did not run the model long enough to match the 14C measurement due to the very high uncertainty in long-term inorganic P cycling and in model initialization. We did test the model for 10,000 years at two other sites with more extreme soil P content, and found out that current inorganic P cycling does not work well in long-term simulation, therefore we have no clue how to initialize the soil mineral P pools, such as primary P pool and secondary P pool, over such a long time. Please find more information in the response to reviewer 3.*

C: We include a new paragraph in the discussion regarding the problem of 14C and inorganic P cycling.

Page17 Line26-Page18 Line6: "Nonetheless, certain caveats of this study and JSM should be discussed. A main challenge is the different simulation times for different purposes. Our results indicated that in the upmost 30 cm of soil, SOM content stabilises after 150 years while in the upmost 1 m SOM stabilises after 1000 years of simulation (Fig.2), regardless of the initial SOM content (Fig.S2). However, with respect to the radiocarbon profile, as indicated by Ahrens et al. (2015), a very long simulation time (13500 years) was required to match both the measured $\Delta^{14}C$ and SOC profiles at a nearby Norway spruce forest site. In our study, a 10000-year simulation time was still not sufficient to match the measured $\Delta^{14}C$ profile, indicating that an even longer simulation time is required. Although JSM is very stable in the long term in term of SOM development and storage, long-term simulation of soil P balance as a result of continuous weathering and occlusion remains a significant challenge (Fig.2, Tab.1). Such a long simulation time is unrealistic for the P cycle due to the unknown conditions of the initial soil P pools and the un-equilibrated soil inorganic P cycling processes (Yang et al., 2014). Although we used a much shorter simulation length in this study, noticeable uncertainties remain due to inorganic P cycling parameters (Tab.2). Additionally, the long simulation time required to match the radiocarbon profiles is also problematic for future coupling to TBMs because these models typically examine centennial time scales. A possible solution is to spin-up radiocarbon (>10000 years) independent of the plant--soil spin-up (1000 years), although this approach needs to be properly tested in the future."

**R: Page 7, lines 30-31: Were there changes in microbial growth rates over the season that could explain changes in microbial N demand? I also would suggest adding some explanation for the large spike in microbial N uptake in November. Is this something to do will autumn litterfall, like a short-term increase in N immobilization due to deposition of a large amount of fresh litter?**

*A: No, we did not find a strong correlation between microbial growth and microbial demand for inorganic N, but of course the total microbial N (organic N + inorganic N) demand is always linear with the growth rate. As we explained in the paper, the microbial inorganic N uptake is largely affected by the N content in DOM.*

*The peak in microbial N uptake in November seems only existing when ECA approach is turned on, indicating that it might be caused by the simulated competition between roots and microbes.*

C: We did not make specific changes regarding this point, as it was already further discussed in the Discussion section "N cycle vs. P cycle". However, we did improve the English in relevant sections to make the result and discussion easier to follow.

**R: Page 8, line 10: What does "TW" mean?**

*A: Removed. It was a comment by co-author we forgot to delete.*

C: Page11 Line17-19: "The simulations showed that microbes outcompeted roots for inorganic P uptake in JSM at all depths."

**R: Page 8, line 27-page 9, line 5: I had trouble following this explanation of the figure, particularly how the potential allocation curves were calculated and how they should be interpreted.**

A: Revised by linking the output in the figure to the variable names and equations. Additionally, a simple description has also been added in the model description. To understand the details of calculation, we would invite the readers to go to the mathematic description in the appendix.

C: Page12 Line9-11: "We compared the enzyme allocation curve of polymeric litter ($Enz_{frac}^{poly}$ in Eq.S17) with three potential allocation curves ($\alpha_{poly}^{X}$ where X stands for C, N, and P, in Eq.S15), which represent cases in which microbes only maximise C, N or P release from depolymerisation."

Page5 Line24-27: "JSM tracks three potential fractions of enzyme allocation, which represent cases in which microbes only maximise depolymerisation release of C, N or P, respectively, and then updates the microbial enzyme allocation fraction by acclimating gradually to the potential fraction of most limiting element (See Sect.S1.5.2)."

**R: Page 9, line 7-8: Microbial N uptake and N losses were not centered around the mean. And there is no Table S4, only S1 and S2.**

*A: Corrected.*

C: Page12 Line25-27: "The interquartile range of outputs (Fig.10) from model sensitivity analysis revealed that all outputs were well centred around the results of the parameterisation of the base scenario (Tab.S2), except microbial inorganic N uptake and N losses."

**R: Page 10, lines 4-9: This seems like an important part of the model structure and results, and should be introduced earlier than the Discussion section. I think this**

**modification to the model should be described in the methods. And since making the parameter depth-dependent makes a difference to the results, it might make sense to include it as a separate set of model simulations (as with the SEAM-off and ECA-off simulations) so its effect could be shown.**

*A: Indeed, the depth-dependent microbial recycling of P is really important for this study site to yield the realistic C:P ratio and Po-to-Pi ratio, and is also what we expect to happen in reality (Rousk and Frey, 2015). We did run a simulation with uniform microbial P recycling along depth but excluded it in the final submission. The reasons to exclude it is that, it is not a standard model feature as SEAM and ECA, which do have theoretical basis. Instead, we suspect that the depth-dependent microbial P recycling should be an emerging model feature if we separate bacteria from fungi.*

*However, we attach the comparison figure here (in the end) and hand it to the editor to decide if it needs to be included or not.*

C: No changes are made yet. Please find the comparison figures below.

**R: Page 10, lines 23-24: At steady state, plant N and P uptake would have to be close to litterfall inputs, unless there were large losses due to leaching or other loss pathways.**

*A: The major reason for not reaching a real equilibrium is, as we stated in the manuscript, the model does not have the feedback from vegetation. That said, we prescribed our litter forcing, and the plant uptake is only determined by the soil conditions regardless of how much plant really requires. As shown in Tab.1, there is no significant loss of N and P from the ecosystem, but N and P are accumulated slowly in the soil due to the fact that 5% of litter fall is accumulated in the soil as SOM.*

C: We have added more results and discussion for the long-term stability of JSM. Please find them in Fig.2, Page9 line15-29, and Page17 Line26-Page18 Line6.

**R: Page 11, Lines 7-8: Is the fact that plants mainly take up N and not mineralized P specific to this ecosystem? In a more P-limited ecosystem, would the results differ?**

*A: We do not know the exact answer to this question. However, in our ongoing work where we run the model with multiple sites along a soil P availability gradient, this pattern still holds true. To our understanding, it is the very different stoichiometry of plant tissue and microbe that yield such a pattern, and it should be even stronger in P poor ecosystem than P rich ecosystem, as indicated by Lang et al. 2017.*

C: No changes are made.

**R: Page 11, line 11-12: The global microbial stoichiometry simulations should be described in the methods.**

*A: Added.*

C: Page8 Line26-Page9 Line2: "To test the effects of different microbial stoichiometry, we ran a *Glob Mic Stoi* scenario in which the global average microbial stoichiometry (42:6:1, Xu

et al., 2013) was used to parameterise the model instead of the observed microbial C:N:P ratio (10.3:0.8:1, Lang et al., 2017). ''

**R: Figure 1: It would be helpful if the notation in this figure matched the notation in the equations in supplementary material.**

*A: Revised.*

C: Please find the changes in Fig.1.

**R: Figure 8: This figure is difficult to understand because there is not a clear explanation of what the different variables mean.**

*A: Revised by linking the variables in legend with their process name and including the order of displayed processes.*

C: Please find the new figure (and caption) of Fig.7

[Figure]

Fig 1. Simulated and observed (a) SOC content, (b) C:N ration in SOM, (c) C:P ratio in SOM, (d) organic P to inorganic P ratio in soil, microbial C, N, and P content ((e) to (g)), and (h) soil bulk density at the study site up to 1m soil depth. Black lines and dots: observations; Color lines and shades: simulated mean values and ranges of standard deviation by different model experiments. The microbial C, N, and P are only measured in top 30cm soil. Simulated means and standard deviations are calculated using data of the last 10 years from the model experiments.

[Figure]

Fig 2. Simulated seasonal and vertical distribution of (a) respiration, (b) net N mineralisation, (c) biochemical P mineralisation, (d) net P mineralisation, (e) microbial inorganic P uptake, (f) plant P uptake, (g) microbial inorganic N uptake, and (h) plant N uptake at the study site up to 1m soil depth. Points represent the mean values and error bars represent the standard deviations, both calculated using data of the last 10 years from the model experiments.

**Responses to Reviewer Comment 3**

**General comments**

R: Yu and coauthors present a conceptually robust model that looks at soil biogeochemical processes that explicitly represents microbial activity and CNP stoichiometry in a vertically resolved model. The work presented here does a very thorough job documenting the model configuration and performance at a well-studied site. What's less clear is why it matters? A few suggestions are described in the specific comments below.

*A: Thanks for the recognition of our effort.*

R: My other major concern with the model is that it doesn't reach steady state equilibrium, instead soil C pools are accumulating at a rate that's roughly 5% of NPP (Table 1). It seems longer spin up times were tried, but since results aren't presented I'm assuming this issue persists, if so, what do soil CNP profiles look like after 10ˆ4 years, do they still match observations well? If the model just has long-term oscillations this may be less of a concern than a constant drift (as I currently understand). The spin up issues, however, seems like a significant issue that has to be addressed if models that more explicitly represent microbial activity and coupled biogeochemical cycles are ever going to be applied in TBMs, as seems to be the aim of this work. The 'lack of plant feedbacks' argument seems unsupported. Moreover, I don't really understand why / how constant 'loss' of P into 'occluded pools affect the C dynamics simulated belowground? This spin-up issue is also one I don't know how to handle in review and my overall assessment of this work. For this reason I'm signing this review and welcome an open conversation with the authors on this concern. I appreciate all the effort that the authors have made do make a very interesting contribution to this line of work- but a model that never really reaches steady state seems very challenging to use for more than short term-studies and sites where the model can be adequately parameterized. This may be the aim of this research group, but it seems unlikely given the introduction, conclusion, and history of strong work from this research group looking at global scale C and nutrient responses for climate change projections.

*A: Thanks for the reviewer to point this out. First of all, in our opinion, the soil system should not reach a real equilibrium due to the fact that soil has to develop from bare soil to certain SOC content, and this accumulation process should not stop as long as the soil is not C-saturated when there are continuous C inputs. However, we do agree in an ideal model simulation, the accumulation rate should be constrained within a very small rate. This is actually the case in the top as well as the near surface subsoil in our model after a few hundred years, while small accumulation continue to take place in the deeper soil. We chose the 200-year simulation length for the manuscript, because the surface soil has already reached equilibrium after 200 years, but the deeper soil continues to accumulate C. In our long-term simulation (5000 yr), the annual accumulation of NPP as C in the soil is only 0.07%, compared with 5% in the 200 yr simulation. There is no evidence of the model application to result in oscillations at longer time-scales, as seen in Fig.1 in which we present the top- and sub-soil C content for the 10000 year simulation. We agree with the reviewer that this has been unclear from the previous version of the manuscript, nor did we include the data in the results. To*

*elaborate this, we will include a new figure in the resubmission to demonstrate how the SOC accumulates in surface and deep soil over a very long time period. We still believe that the results of our study can reasonably be interpreted, because the top 30 cm showed near equilibrium conditions already after 200 years (demonstrated in Fig.S1 and S2).*

*A general issue with the development of stand-alone nutrient enabled soil biogeochemical models is that the assumed plant uptake demand does not adequately reflect long-term soils development. When the model was ran for a very long time (e.g. 10,000 years), there were some cases in which the primary P in surface layers got depleted and a large fraction of the  sorbed P got occluded. While microbes detect this change and as a result levels down its biomass because it takes up less P, the root biomass and associated plant P uptake in our model is prescribed at the level of mature healthy forest. That said, the root biomass does not change under P limited growth condition, nor does the root distribution over soil layers change. The lack of the phosphorus-root growth feedback implies that under such conditions fine roots become more competitive than microbes in taking up inorganic P, and there is always living roots trying to take up P even if they only take up little P for a very long time. The inorganic P cycling problem is a common problem for the community of terrestrial biosphere modelers, especially at very P-poor ecosystems.*

C: We have added more results and discussion for the long-term stability of JSM. Please find them in Fig.2, Tab.1, Page9 line15-29, and Page17 Line26-Page18 Line6.

**Specific comments**

**R: In my opinion there's a bit too much emphasis in the introduction in playing up the novelty of this work. This is not the first model to think about vertical resolution, microbes, nutrients or ECA. It may be the first to do all these together, which can be stated, but then move on. The current review of the literature is nice, but I'd encourage the authors to avoid language that's unnecessarily dismissive of previous work. To address my first issue of 'why this work matters' I can three of three options to consider:**

*A: We are grateful for all the previous work that makes this model possible and we do realize our wording has caused misunderstandings. We'll carefully rephrase them in the resubmission.*

C: We have made several changes to acknowledge other researchers' work in the introduction, such as Page 2 Line 26, Page2 Line29-32, Page 3 Line21-23.

1.  **Idealized experiments: While the justification for including nutrients and microbial feedbacks in a model like the Jena soil model is well established in the abstract and first paragraphs of the introduction I also fear it sets up somewhat unrealistic expectations for readers. Notably, none of the results presented illustrate how the model may respond to environmental perturbations. I'm not suggesting these have to be compared to results, but instead simple idealized experiments that illustrate how the different model configurations respond to increases in litterfall inputs, root exudates, warming, or changes in precipitation.**

A: We really appreciate the suggestions by the reviewer for more interesting model experiments, and we are also interested in carrying out such experiments in the future. However, performing such model experiments themselves in a meaningful way would require substantial additional model evaluation and discussion to discuss whether the simulated feedbacks are commensurate with current understanding. Because established model benchmarks do not exist for this model behaviour, this would require an in-depth discussion of the available observations, which in our opinion is beyond the scope of a model description paper. Simply showing sensitivity study without comparing these to suitable observations would be fairly meaningless.

C: no changes are made regarding this point.

2. **Model validation: Alternatively, it seems lots of data were needed to initialize the model. This is fine for development, but how well does the model do simulating other sites? Are there other well studied sites that can be used for independent model validation? I realize this potentially an objective for future work, but it seems like typical activity for model development papers (especially in GMD) that would help illustrate the broader generalizability of the approach outlined here?**

A: Thanks again for the suggestions. Simulation on multiple sites, e.g. a gradient study involves specific biogeochemistry scientific questions to be addressed, and we believe that this is beyond the scope of this paper. Simply showing that the model could be calibrated in other sites will not give much additional value to this paper.

C: no changes are made regarding this point.

3. **Sensitivity analysis: A third alternative would be to consider illustrating model sensitivities to initial conditions? Much like the idealized experiment suggestion (above), I kept finding myself wondering how sensitive the model behaves to initial conditions that are being input to the model (e.g. litterfall and microbial stoichiometry, soil texture / mineralogy, water fluxes and temperature profiles). The parameter sensitivity analysis is nice, what about other assumptions that are being made regarding inputs to what seems like a highly parameterized model? This would open up the discussion for consideration of how to run JSM in regional or global simulations (clearly the intent), where we have less certainty of how the define these characteristics (especially with multiple elements and with depth).**

A: Thanks for the comment.  We did test how model performs under different initial conditions, and we have also done some other experiments, such as how the model responds to different microbial carbon-use-efficiency and nutrient-use-efficiencies, plant/microbe uptake rates of mineral nutrients, DOM uptake rates etc.. The reasons for not showing all of them are very similar as the ones for previous two: first, there is a limit on how much we can try to include in a model description paper; second, some of the experiments are very interesting topic to formulate new studies, and we don't want to dilute the importance of them by including them into this paper.

We will include the model results under different initial conditions in the resubmission, and discuss it together with the spin-up, equilibrium state, and stability issues.

C: we have included the model test of different initial conditions: method description at Page 8 Line28-Page9 Line2, result displayed in Fig.S2 and presented at Page 9 Line16-19 and discussed at Page 17 Line28-29.

**R: The authors have actually done #3 with the microbial stoichiometry section that squeezed into the discussion. Maybe the most direct path forward to satisfy this concern would be to actually flush out these findings in the methods and results (see technical comment below).**

*A: Thanks for the recognition and we will make it more visible in the resubmission.*

C: Page8 Line26-28: "To test the effects of different microbial stoichiometry, we ran a *Glob Mic Stoi* scenario in which the global average microbial stoichiometry (42:6:1, Xu et al., 2013) was used to parameterise the model instead of the observed microbial C:N:P ratio (10.3:0.8:1, Lang et al., 2017). "

**Technical corrections**

**R: Page 2, Line 10, I might include Lehmann and Kleber 2015 here.**

*A: Thanks, included.*

C: Page2 Line23-24: "…the nutrient immobilisation and physical stabilisation of organic matter through organo-mineral association (Schmidt et al., 2011; Lehmann and Kleber, 2015)."

**R: Page 2, Line 11, Vertically resolved models are becoming more common (McGuire et al. 2018)**

*A: Thanks, included.*

C: Page2 Line25-26: "… thus ignoring the strong variance of SOM cycling within a soil profile (Koven et al., 2013; Arora et al., 2013;McGuire et al., 2018)."

**R: Page 2, Line 18, I'm not sure the assertion (made here and in the following paragraph) that microbial explicit models don't represent coupled biogeochemical cycles is accurate (Averill & Waring 2017; Schimel & Weintraub 2003; Sistla et al. 2014; Sulman et al. 2017, 2019).**

*A: Thanks. We will carefully revisit related literature and revise the introduction, as both reviewer 2 and 3 have pointed out the problem.*

C: Page3 Line19-24: "… accounting for the N cycle, enzymatic biosynthesis and rhizosphere priming (Abramoff et al., 2017; Sulman et al., 2017; Huang et al., 2018; Sulman et al., 2019) using certain novel approaches."

**R: Page 3, Line 4. I'm pretty sure the ECA approach is applied in E3SM land model, which I wouldn't call a prototype model.**

*A: Corrected.*

C: Page3 Line33-35: "ECA has also been applied to resolve mineral nutrient sink (plant–microbe uptake or mineral adsorption) competitions in other modelling studies (Zhu et al., 2016, 2017)."

**R: Page 3, line 16. & Page 4. Where's section 5?**

*A: Should be Sect. S2, corrected.*

C: Page5 Line29: "…, following the ECA approach (See Sect.S2.2)."

**R: Methods. I know COMISSION already has radiocarbon, but should there be any focus on documenting how JSM implements radiocarbon in the text or appendix?**

*A: We will include the 14C initialization in the model protocol section. But we don't intend to include the radiocarbon in the supplementary material since they are not the new development of this paper.*

*We will clearly state it in the revision that the model explicitly traces 14C, see Ahrens et al. 2015 and Thum et al. 2019, and the 14C values of litter forcing were generated using QUINCY, see Thum et al. 2019.*

C: Page5 Line17: "JSM explicitly traces $^{13}$C, $^{14}$C and $^{15}$N following Ahrens et al. (2015) and Thum et al. (2019)."

Page7 Line16: "… and vertically resolved litterfall that includes $^{14}$C values) …"

**R: Page 7 and Fig 2 the model calculates its own bulk density?! That's pretty interesting, should this be described in the methods?**

*A: We will include it in the model description, together with the descriptions of some other processes, as mentioned in the response to reviewer 2.*

C: Page6 Line1-2: "JSM also considers the effects of SOM content to correct bulk density (Sect.S3), which in turn affects other processes such as organic matter (Eq.S7) and phosphate (Eq.S25) sorption."

**R: Page 7, Line 10-15. It seems odd to jump from presentation of Fig 2 to 7. Should the display items reflect the order that information is covered in the text?**

**Throughout, display items should be numbered in the order they are introduced in the text.**

*A: Thanks for pointing out the problem. We will revise the order of our displayed items in the resubmission.*

C: all the displayed items in the manuscript are re-numbered in their order of appearances.

**R: Fig 8 and Table 1 are never referenced in the results, should they be? I'd prefer these display items not be first introduced in the discussion of the findings of this study.**

*A: Thanks for pointing it out. We will reorganise the results and discussions according to the order of display items. However, we do think these findings are interesting enough given the fact it is a model description paper. More elaboration can be found in the response to comments of "Discussion".*

C: Page9 Line22: "… , but the complete soil profile had not yet reached a steady state (Tab.1) …"

Page11 Line5-7: "The sources and sinks of soluble inorganic N and P also show very different patterns (Fig.7). The main source and sink for inorganic N in solution are gross mineralisation and plant uptake ofNH4, respectively; whereas for P, microbial uptake is the main sink and biomineralisation is a larger source than gross mineralisation in each scenario."

**R: Fig. 7 Bottom panels of should be % modern. I also couldn't help but notice that you just have 14C data for the site. Why not run the model for longer and show result, or put the radiocarbon observations up on the plot shown here even if they're just illustrative for 14MOC (which should be most of what makes up the bulk 14C values at depth? Wait, the 14C data are presented in the SI (page 7, line 20- sorry I'm on a plane and don't have access to the SI material). It seems this would be a powerful constraint for the model to try and hit (and should be included in the main text). I'm struck that we can learn a good deal about the model, even if the model is not able to match radiocarbon profiles! If longer spin-up runs have already been done I can't think of any reason not to compare results to observations where they are available.**

*A: Since this study does not involve any development of radiocarbon calculation, we did not focus on presenting the 14C results. The main message of the 14C results in this paper is that, the inclusion of N and P cycling and other processes do not affect the capacity of the carbon core of JSM (i.e. COMISSION model) to capture/approach the soil profile radiocarbon.*

*Admittedly, we have stated in the paper that due to the uncertainty in initialization and P cycling processes, the model will have P depletion problem in the long-term simulation (>10,000 years). As a demonstration, we show the change of non-occlude inorganic P for 10,000 year below (Fig.2). The P content in top- and sub-soil fluctuates before 2500 years due to the combined effects of transport and immobilization/mineralization, but after that both of them decrease continuously. We did reach P depletion in other long-term (10,000 year) tests during the calibration processes although this one is not yet there.*

*We agree that a long simulation time is the perquisite to hit the 14C soil profile, but in order to run the model stably for such a long time, we might need to switch off some inorganic P cycling processes in the spin-up. We will discuss about this more in detail in the resubmission.*

C: Page17 Line26-Page18 Line6: "Nonetheless, certain caveats of this study and JSM should be discussed. A main challenge is the different simulation times for different purposes. Our results indicated that in the upmost 30 cm of soil, SOM content stabilises after 150 years while in the upmost 1 m SOM stabilises after 1000 years of simulation (Fig.2), regardless of

the initial SOM content (Fig.S2). However, with respect to the radiocarbon profile, as indicated by Ahrens et al. (2015), a very long simulation time (13500 years) was required to match both the measured $\Delta^{14}C$ and SOC profiles at a nearby Norway spruce forest site. In our study, a 10000-yearsimulation time was still not sufficient to match the measured $\Delta^{14}C$ profile, indicating that an even longer simulation time is required. Although JSM is very stable in the long term in term of SOM development and storage, long-term simulation of soil P balance as a result of continuous weathering and occlusion remains a significant challenge (Fig.2, Tab.1). Such a long simulation time is unrealistic for the P cycle due to the unknown conditions of the initial soil P pools and the un-equilibrated soil30inorganic P cycling processes (Yang et al., 2014b). Although we used a much shorter simulation length in this study, noticeable uncertainties remain due to inorganic P cycling parameters (Tab.2). Additionally, the long simulation time required to match the radiocarbon profiles is also problematic for future coupling to TBMs because these models typically examine centennial time scales. A possible solution is to spin-up radiocarbon (>10000 years) independent of the plant–soil spin-up (1000 years), although this approach needs to be properly tested in the future."

**R: Figs 3-4, Page 7. From the text it sounds like there are observations of soil nutrient transformation (at least N mineralization). If so, can these be included on the appropriate panels, or am I misunderstood?**

*A: Sorry for the confusion, but we don't have observed nutrient fluxes that can be comparable to our simulations.*

C: no changes are made.

**R: Page 8, line 10, what is TW in JSM? Section 3.2. Is the strong microbial competition for P (and not N) caused by the C:N:P ratios that are prescribed for the site (and notably skewed).**

*A: The content within the bracket in line 10 is a co-author's comment which should have been removed.*

*The reviewer 2 also has similar concern about the strong microbial P competition of the site, but as what we have seen in simulations of other sites (for another study) it is a consistent pattern in all sites. However, the C:N:P ratios of this study site is not far from other sites we have (Lang et al. 2017), but very far from the global average value. The scenario using global average microbial stoichiometry also shows that microbe outcompetes roots, but not as strong as the base scenario.*

C: Page11 Line17-19: "The simulations showed that microbes outcompeted roots for inorganic P uptake in JSM at all depths."

**R: Page 8, line 24, the difference among models mentioned here regarding depth profiles of N-mineralization is not obvious, at least to my eye. Regardless, avoid using 'significant' when no statistical results are presented.**

A: Thanks for the suggestion. It will be corrected in the resubmission.

C: Page12 Line3-4: "…, but decrease in net N mineralisation with soil depth is marginally stronger (Fig.5). "

**Page 8, line 27, it's not clear from the methods how the actual and potential enzyme allocation curves are being calculated from the methods, or did I miss this description. I'm also still hung up on how or why this is being done if the model doesn't explicitly represent enzymes (by the way this decision not to explicitly represent enzymes makes sense to me from a purely practical / numeric standpoint)**

*A: The detailed processes descriptions are presented in the supplementary material. For the enzyme allocation, we made an assumption that the total enzyme is always proportional to the microbial biomass and used the enzyme richness in the Michaelis-menton equation. Therefore we did implicitly model the enzyme production, and explicitly model the enzyme allocation. We will clarify this in the model description in the main manuscript.*

C: Page5 Line24-27: "JSM tracks three potential fractions of enzyme allocation, which represent cases in which microbes only maximise depolymerisation release of C, N or P, respectively, and then updates the microbial enzyme allocation fraction by acclimating gradually to the potential fraction of most limiting element (See Sect.S1.5.2)."

**R: Page 8 line 33, if P depolymerization is completely demand driven why is microbial P uptake so much lower in the ECA-off simulations (Fig 4a)? I thought these were supposed to be the 'demand based' simulations (methods)? Please clarify.**

*A: In the ECA approach, we do not calculate the demand, but the potential uptake depends not only on the uptake capacity per carbon roots/microbes, but also on the biomass of roots and microbes. The ECA approach mainly regulates the competition of uptake capacity per carbon, but eventually the total uptake still depends on the microbial biomass. That is why it looks like "demand-based" simulation.*

*To simply explain what happened when we turned off ECA: we initialize all the scenarios the same, but the microbes take up less P per biomass carbon than the base scenario, therefore the microbes develop less biomass than the base scenario. Both the lower microbial biomass and lower uptake capacity per unit carbon in the ECA-off scenario has caused the much lower microbial P uptake than the base scenario.*

C: No changes are made.

**R: Page 9, line 5. Reference Fig 7 here?**

*A: Corrected.*

C: Page12 Line23: "…, resulting a systematic difference in the radiocarbon profiles between the two scenarios (Fig.4)."

**R: Page 9, Line 25 these values are for soil stoichiometry? Also, what are N:P ratios for soils? Finally, to my eye it looks like the model may overestimate observed soil C:N**

**ratios in upper soil horizons (Fig 2). Regardless, it's likely helpful to point to this display item to support claims made about soil C pools and stoichiometry made here.**

*A: Thanks for pointing out the problem. It is the soil stoichiometry we are discussed here, and the C:N ratio in the O-A horizon is indeed overestimated by the model. We will include it in the discussion when we resubmit.*

C: Page13 Line25-27: "Slight overestimation of the modelled soil C:N ratio in the first layer (Fig.3) is probably due to the higher C:N ratio (52) of leaf litter inputs than the observed one (41.7)."

Page13 Line16: "The observed SOM C:N ratio (19.5)and C:P ratio (348) in the first model layer …"

**R: Discussion: I have to admit I haven't thought much about the dynamics driving declines in soil C:P ratios with depth, nor am I very familiar with this literature. For everything the model is doing here, this text strikes me as an odd choice to highlight at the beginning of the discussion. That said, it. Is interesting. One detail I don't really follow is that to capture observations it seems like the P recycling term in the model has to be greatly reduced in model. It doesn't seem to logically follow that the community somehow shifts to 'nutrient rich' community that's also has lower nutrient use efficiency? Instead I think the findings of Rousk and Frey suggest that substrate quality determines the microbial communities in forest soils, but doesn't speak much to vertical distribution of microbes (or their stoichiometry) being. Discussed here?**

*A: Thanks for the comment. We think this finding is interesting and new, and should be stated early in the discussion. First of all, the soil stoichiometry is a rarely discussed topic in the modeling community, and the fact that C:P ratio decreases much faster than C:N ratio with depth is also very interesting for us. Besides, we only have observations for the soil stocks but not flux, so it is natural for us to start with the finding that we saw in the soil stocks. However, as all the reviewers are concerning about the model spin-up, stability/equilibrium state, we will also include this topic in the first part of the discussion.*

*For the second part of the question, we found that the model has to be tuned in a way that the microbial residue becomes P-poor in the surface layer to reproduce the C:P depth profile. To us it means the microbes need to be more dominated by fungi in the surface soil, and it agrees with what Rousk and Frey (2015) presented in their results (Table 2) that organic layer has higher fungi:bacterial ratio than mineral soil. It also agrees with one of their conclusions that more litter input will lead to higher fungi:bacterial ratio. Although they did not mention soil depth specifically, it is an obvious fact that litter input to soil decreases with soil depth.*

C: No specific changes are made.

**R: Table 1 should include soil C, N, & P pools of the model after spin up, as it's hard to assess total pool sizes from figures.**

*A: We only looked at the last 10 years' pool size change of the simulation. We will also include a new figure to demonstrate how the total pool size changes over 10000 years.*

C: see the new figure, Fig.2, and relevant discussions about it. Readers can easily see the change of pool size within 10000 years' time in the new figure.

**R: Fig 8 can colors of processes in the legend match the order they are displayed on the figure. As currently presented it's not easy for readers to interpret the figure. Introduction of the microbial stoichiometry part of the discussion seems like a nice sensitivity test of the model, but I don't like this being squeezed into the discussion and SI. Why not at least justify this experiment in the methods and describe findings in the results before discussing the findings? (It also likely makes sense to keep the figures in SI).**

*A: Thanks for the suggestion. We will include the microbial stoichiometry scenario in the methods section.*

C: we improved the figure to better convey the information, and also include a short paragraph in Results section to help readers.

Page11 Line5-7: "The sources and sinks of soluble inorganic N and P also show very different patterns (Fig.7). The main source and sink for inorganic N in solution are gross mineralisation and plant uptake ofNH4, respectively; whereas for P, microbial uptake is the main sink and biomineralisation is a larger source than gross mineralisation in each scenario."

**R: Page 12, line 25. What observations are the model able to reproduce? Can they be illustrated on the display items (* that also should be referenced here)?**

*A: We will rephrase the sentence to be more precise and relate to the display items.*

C: Page17 Line9-11: "JSM demonstrated a capacity to reproduce the vertical patterns of soil stocks (Fig.3) and to satisfactorily produce both vertical and seasonal patterns of biogeochemical fluxes (Fig.5 and 6)."

**R: Page 12, line 28, why not cite a commission paper that's already published**

*A: We have taken the carbon cycling framework of the most recent version of COMISSION (Ahrens et al. 2019, in review), which is already different from the published version (Ahrens et al. 2015). We will update the reference once it is accepted.*

C: No specific changes are made.

[revised manuscript text omitted]

---

## Author Response (AR3)

**R: reviewer's comment**

*A: authors' response*

C: changes made in the manuscript (the tracked-change version)

**Responses to Reviewer Comment**

**R: P2, Line 11, replace "physical stabilisation" with "physical-chemical stabilization".**
*A: Thanks. Changed.*
C: Page 2, Line12: "…as well as the nutrient immobilisation and physical-chemical stabilisation of organic matter …"

**R: P8, Line 30, I think the simulated d14C only agreed with observation "qualitatively", therefore it should be acknowledged here.**
*A: Thanks. Changed.*
C: Page 8, Line30: "The simulated radiocarbon (Δ14C) profile qualitatively agreed with observed one …"

**R: Also, the following paper is another recent effort in concurrent modeling of CNP cycles in ESM land models.**
**Zhu Q, Riley WJ, Tang JY, Collier N, Hoffman FM, Yang XJ, Bisht G (2019) Representing Nitrogen, Phosphorus, and Carbon Interactions in the E3SM Land Model: Development and Global Benchmarking. J Adv Model Earth Sy 11(7): 2238-2258**
*A: Thanks. Added.*
C: Page 1 Line22, and Page 3 Line 10

**R: Finally, one cosmetic suggestion, maybe the axes ticks and labels of the figures can be larger for a better visual.**
*A:Thanks, modified.*
C: The font of axes ticks and labels of figure 2 to figure 10 in the main text has all been increased.

[revised manuscript text omitted]
^{C:N}_{mic}$ | -0.99 | $v^{res}_{max,depoly}$ | -0.94 | $\eta_{C,sol\to dom}$ | 0.84 | $\chi^{N:P}_{mic}$ | 0.40 | $\eta_{C,wl\to poly}$ | 0.38 |
| Total SOP | $\chi^{C:N}_{mic}$ | -0.97 | $\chi^{N:P}_{mic}$ | -0.97 | $v_{max,biomin}$ | -0.84 | $\eta^{P}_{res\to dom}$ | -0.78 | $\frac{1}{\tau_{mic}}$ | 0.74 |
| Total SIP | $k_{weath}$ | -0.58 | $\eta^{P}_{res\to dom}$ | 0.57 | $v_{max,biomin}$ | 0.47 | $\chi^{N:P}_{mic}$ | 0.45 | $\eta^{P}_{res\to dom}$ | -0.42 |
| Microbial C | $\frac{1}{\tau_{mic}}$ | -0.98 | $\eta_{C,sol\to dom}$ | 0.86 | $\chi^{C:N}_{mic}$ | 0.68 | $\chi^{N:P}_{mic}$ | 0.67 | $\eta_{C,wl\to poly}$ | 0.67 |
| Microbial N | $\frac{1}{\tau_{mic}}$ | -0.97 | $\chi^{C:N}_{mic}$ | -0.95 | $\eta_{C,sol\to dom}$ | 0.83 | $\chi^{N:P}_{mic}$ | 0.63 | $\eta_{C,wl\to poly}$ | 0.62 |
| Microbial P | $\frac{1}{\tau_{mic}}$ | -0.96 | $\chi^{N:P}_{mic}$ | -0.94 | $\chi^{C:N}_{mic}$ | -0.93 | $\eta_{C,sol\to dom}$ | 0.79 | $\eta_{C,wl\to poly}$ | 0.55 |
| Respiration | $\frac{1}{\tau_{mic}}$ | -0.71 | $\chi^{N:P}_{mic}$ | 0.69 | $\chi^{C:N}_{mic}$ | 0.65 | $v^{res}_{max,depoly}$ | 0.45 | $mic^{min}_{cue}$ | -0.37 |
| Net N mineralisation | $\chi^{C:N}_{mic}$ | 0.97 | $v^{res}_{max,depoly}$ | 0.65 | $mic^{min}_{cue}$ | -0.40 | $\eta_{C,sol\to dom}$ | -0.32 | $\frac{1}{\tau_{mic}}$ | -0.29 |
| Microbial N uptake | $mic_{nue}$ | -0.98 | $\chi^{C:N}_{mic}$ | -0.90 | $\eta_{C,sol\to dom}$ | 0.75 | $\eta_{C,wl\to poly}$ | 0.38 | $\chi^{N:P}_{mic}$ | 0.21 |
| Net P mineralisation | $\eta^{P}_{res\to dom}$ | 0.94 | $\chi^{N:P}_{mic}$ | 0.84 | $\chi^{C:N}_{mic}$ | 0.84 | $\eta_{C,sol\to dom}$ | -0.67 | $\eta_{C,wl\to poly}$ | -0.53 |
| P Biomineralisation | $\eta^{P}_{res\to dom}$ | -0.94 | $\chi^{N:P}_{mic}$ | -0.85 | $\chi^{C:N}_{mic}$ | -0.84 | $\eta_{C,sol\to dom}$ | 0.67 | $\eta_{C,wl\to poly}$ | 0.54 |
| Microbial P uptake | $mic_{pue}$ | -0.91 | $\chi^{N:P}_{mic}$ | -0.90 | $\chi^{C:N}_{mic}$ | -0.89 | $\eta^{P}_{res\to dom}$ | -0.85 | $\eta_{C,sol\to dom}$ | 0.70 |
| N Losses | $\chi^{N:P}_{mic}$ | 0.72 | $\frac{1}{\tau_{mic}}$ | -0.72 | $\chi^{C:N}_{mic}$ | 0.67 | $v^{dom}_{max,upt}$ | 0.41 | $v_{max,biomin}$ | 0.35 |
| P Losses | $v^{dom}_{max,upt}$ | 0.22 | $mic_{pue}$ | 0.15 | $mic^{min}_{cue}$ | -0.14 | $\eta^{P}_{res\to dom}$ | -0.11 | $k^{P}_{enz,mic}$ | -0.55 |
| OVI | $\chi^{C:N}_{mic}$ | 0.73 | $\chi^{N:P}_{mic}$ | 0.57 | $\frac{1}{\tau_{mic}}$ | 0.47 | $\eta_{C,sol\to dom}$ | 0.42 | $\eta^{P}_{res\to dom}$ | 0.35 |